# MCT1-dependent energetic failure and neuroinflammation underlie optic nerve degeneration in Wolfram syndrome mice

Greta Rossi[1], Gabriele Ordazzo[1], Niccolò N Vanni[1], Valerio Castoldi[1,2], Angelo Iannielli[1,3], Dario Di Silvestre[4], Edoardo Bellini[1], Letizia Bernardo[4], Serena G Giannelli[1], Mirko Luoni[1,3], Sharon Muggeo[1], Letizia Leocani[1,2], PierLuigi Mauri[4], Vania Broccoli[1,3]*

[1]Division of Neuroscience, San Raffaele Scientific Institute, Milano, Italy; [2]Experimental Neurophysiology Unit, Institute of Experimental Neurology (INSPE), San Raffaele Scientific Institute, Milan, Italy; [3]National Research Council of Italy, Institute of Neuroscience, Milano, Italy; [4]National Research Council of Italy, Institute of Technologies in Biomedicine, Milan, Italy

**Abstract** Wolfram syndrome 1 (WS1) is a rare genetic disorder caused by mutations in the *WFS1* gene leading to a wide spectrum of clinical dysfunctions, among which blindness, diabetes, and neurological deficits are the most prominent. *WFS1* encodes for the endoplasmic reticulum (ER) resident transmembrane protein wolframin with multiple functions in ER processes. However, the *WFS1*-dependent etiopathology in retinal cells is unknown. Herein, we showed that *Wfs1* mutant mice developed early retinal electrophysiological impairments followed by marked visual loss. Interestingly, axons and myelin disruption in the optic nerve preceded the degeneration of the retinal ganglion cell bodies in the retina. Transcriptomics at pre-degenerative stage revealed the STAT3-dependent activation of proinflammatory glial markers with reduction of the homeostatic and pro-survival factors glutamine synthetase and BDNF. Furthermore, label-free comparative proteomics identified a significant reduction of the monocarboxylate transport isoform 1 (MCT1) and its partner basigin that are highly enriched on retinal glia and myelin-forming oligodendrocytes in optic nerve together with wolframin. Loss of MCT1 caused a failure in lactate transfer from glial to neuronal cell bodies and axons leading to a chronic hypometabolic state. Thus, this bioenergetic impairment is occurring concurrently both within the axonal regions and cell bodies of the retinal ganglion cells, selectively endangering their survival while impacting less on other retinal cells. This metabolic dysfunction occurs months before the frank RGC degeneration suggesting an extended time-window for intervening with new therapeutic strategies focused on boosting retinal and optic nerve bioenergetics in WS1.

*For correspondence:
broccoli.vania@hsr.it

Competing interest: The authors declare that no competing interests exist.

## Editor's evaluation

The primary goal of this paper is to characterize retinal dysfunction and retinal ganglion cell degeneration in the Wfs1exon8del murine model of Wolfram Syndrome 1. The study provides fundamental insight into the timelines of degeneration as well as valuable transcriptomic and proteomic datasets. The methodologies performed are rigorous and the conclusions reached are generally well supported by the data. The results of this study are highly relevant for molecular mechanisms in Wolfram Syndrome 1 and are of potential interest to scientists interested in oligodendrocyte and neuron communication.

## Introduction

Wolfram syndrome 1 (WS1) is a rare and multisystemic genetic disease. In most cases the first clinical sign is the development of non-autoimmune diabetes mellitus (DM) during infancy followed by optic atrophy (OA) which leads to progressive visual loss in adolescence progressing relentlessly to blindness in adulthood (*Wolfram and Wagener, 1938*; *Barrett et al., 1995*). While DM symptoms are clinically treatable with antidiabetic therapies, interventions to arrest or delay the loss of sight are lacking (*Urano, 2016*). Although concurrent DM and OA are pathognomonic for WS1, patients might develop additional serious neurological illnesses, most commonly hearing loss, ataxia, epilepsy, and peripheral neuropathy (*Minton et al., 2003*; *Chaussenot et al., 2011*). Serious psychiatric disturbances have also been described in WS1 patients including psychosis, episodes of severe depression, and impulsive/aggressive behavior (*Swift et al., 1990*; *Swift and Swift, 2005*). OA is caused by the selective loss of retinal ganglion cells (RGCs) and their axons in the optic nerve with a pattern of degeneration that mainly interests the central part of the optic tract rather than the periphery and is accompanied by different degrees of demyelination (*Zmyslowska et al., 2019*; *Barboni et al., 2022*). Noteworthy, neuroimaging studies revealed diffuse alterations in the gray and white matters of the brain with significant loss of volume of the brainstem and microstructure abnormalities and signs of degeneration of myelin (*Lugar et al., 2016*; *Samara et al., 2019*). The gene mutated in WS1 is *WFS1* which encodes for a multi-transmembrane protein and resident in the endoplasmic reticulum (ER) named wolframin (*Inoue et al., 1998*; *Strom et al., 1998*). Mounting evidence suggests that *WFS1* gene loss induces ER stress-mediated apoptosis of insulin producing pancreatic β-cells with disruption of cellular calcium homeostasis (*Fonseca et al., 2005*; *Fonseca et al., 2010*; *Riggs et al., 2005*; *Yamada et al., 2006*). In particular, wolframin is able to increase the $Ca^{2+}$ uptake within the ER, partially through the regulation of the sarco/endoplasmic reticulum $Ca^{2+}$-ATPase SERCA pump (*Zatyka et al., 2015*). As a consequence, wolframin loss results in the $Ca^{2+}$ leakage from the ER with a simultaneous increase of its cytosolic concentration, activation of the $Ca^{2+}$-dependent protease calpain which, in turn, activates programmed apoptosis (*Lu et al., 2014*; *Takei et al., 2006*). Moreover, recent findings have suggested that wolframin is required for the proper release of both insulin and neurotransmitters from the ER and their delivery to the cell periphery through assisted vesicle trafficking (*Wang et al., 2021*). Increasing reports have described dysfunctional ER-mitochondria communication caused by wolframin loss. In fact, *WFS1* deficiency deranges the inositol 1,4,5-trisphosphate receptor (IP3R)-mediated release by destabilizing the NCS1 protein and, thereby, leading to reduced $Ca^{2+}$ uptake by the mitochondria with consequent functional impairments (*Schlecker et al., 2006*; *Cagalinec et al., 2016*; *Angebault et al., 2018*). Moreover, a recent study unveiled an enriched localization of wolframin at the MAM sites and in WS1 patient fibroblasts, wolframin deficiency correlated with MAMs loss and, therefore, with defective $Ca^{2+}$ transfer from the ER to the mitochondria and their related dysfunctions (*La Morgia et al., 2020*). Remarkably, pharmacological activation of the sigma-1 receptor (S1R), an ER-resident protein involved in calcium ion transfer, was shown to counteract the functional alterations of MAMs due to wolframin deficiency (*Crouzier et al., 2022a*). Altogether, these results highlight mitochondrial dysfunctions as a relevant pathogenetic event triggered by wolframin and leading to cell death. However, most of these findings were obtained using cell lines and patients' fibroblasts, and, therefore whether these pathogenetic alterations have the same relevance in neural cells and represent the main and only cause of neurodegeneration remain to be further investigated. Different *Wsf1* targeted mutant mouse and rat lines have been generated and showed to recapitulate the main cardinal pathological traits of the WS1 patients. Furthermore, zebrafish mutant lines for the two homologue genes wfs1a and wfs1b have been recently produced showing retinal abnormalities and behavioral visual dysfunctions associated with ER stress and impaired UPR response in both strains (*Cairns et al., 2021*; *Crouzier et al., 2022b*). Between the mouse models, the neurological and behavioral alterations of the *Wfs1*exon8del targeted mice have been extensively characterized showing diabetic glucose intolerance and anxious-like behaviors upon stressful environment exposure (*Kato et al., 2008*; *Luuk et al., 2008*). Visual impairment in these mice is less characterized with only a histopathological study that reported some thinning of the retinal tissue with lower retinal thickness/longitudinal diameter ratio in 4 months old *Wfs1*-deficient mice (*Waszczykowska et al., 2020*). More recently, a mutant rat line was obtained through *Wfs1* exon 5 disruption showing the hallmarks of WS1, with diabetes, optic atrophy, and neurodegeneration (*Plaas et al., 2017*). The increase of ER stress levels was detected in mutant retina, brainstem and pancreas, coupled with a volumetric decrease in all these districts.

These features made it very useful to test the efficacy of the pharmacological treatments (*Plaas et al., 2017*). Prolonged administration of the glucagon like peptide 1 receptor (GLP1-R) agonists have demonstrated beneficial effects in preventing diabetes, excitotoxicity, and ER stress in aged rats and delaying disease progression (*Toots et al., 2018*; *Seppa et al., 2019*). Very recently the synergistic effects of GLP1-R agonist and the brain-derived neurotrophic factor mimetic 7,8-DHF co-treatment have shown neuroprotective capacity on rats' visual pathway rescuing visual activity (*Seppa et al., 2021*). Interestingly, chronic ER stress and autophagy alterations caused by *Wfs1* loss results in the acceleration of Tau protein aggregation and relative pathology in mouse hippocampus suggesting a novel mechanism by which wolframin deficiency can lead to neurodegeneration (*Delpech et al., 2021*; *Chen et al., 2022*). Thus, rodent models of *Wfs1* deficiency offers pivotal systems where to dissect and further validate the pathogenetic basis of this disease. Herein, we employed the *Wfs1*exon8del mutant mice to better characterize the progressive impairments of visual physiology and activity over time and identify novel molecular and biochemical alterations by genome-wide transcriptomics and proteomics on isolated retinal tissues. These targeted analyses reveal neuroinflammation and bioenergetic substrate loss as plausible causes for the selective vulnerability of RGCs and their degeneration in WS1 mice.

## Results

### Progressive visual impairments in *Wfs1* mutant mice

To determine RGCs functionality and visual signal conduction to the brain in *Wfs1* mutant mice, we exploited whole field photopic electroretinogram (pERG), which measures action potentials produced by the retina when it is stimulated by light of adequate intensity and it is the composite of electrical activity from the photoreceptors, inner retina, and RGCs. In photopic conditions, the negative response (PhNR) wave represents the RGC spiking activity originating from light-adapted cones and allows the detection of RGC-specific alterations (*Gotoh et al., 2004*; *Kinoshita et al., 2016*). Whilst pERG measurements were comparable between wild-type and *Wfs1* mutant mice at 2 months of age (*WT*: 68±5; *Wfs1*-KO: 72±6; p<0.05), a prolonged implicit time, which refers to the interval between the stimulation and the peak of the negative response, was detected in 4 months old *Wfs1*-deficient animals (*WT*: 76±5; *Wfs1*-KO: 82±6; p<0.05) (*Figure 1A*). This early indication of RGC abnormalities became increasingly more pronounced in 8 (*WT*: 73 ± 6; *Wfs1*-KO: 83 ± 9; *p* < 0.05) and 12 (*WT*: 76 ± 5; *Wfs1*-KO: 85 ± 5; *p* < 0.0001) months old mutant mice, with the latter group that showed, beside a delayed response , also a significant PhNR amplitude decrease (*WT*: 11±2; *Wfs1*-KO: 8±2, p<0.0001) at the latest time period (*Figure 1A*). To further deepen our neurophysiological evaluation, we sought to investigate light-induced electric conduction from the retina to the brain visual cortex by flash visual evoked potential (fVEP) recordings. These measurements represent a direct readout of the optic nerve functional activity by providing two important parameters: the latency, which measures the propagation speed of the visual stimulus along the nerve fibers until its central detection; the amplitude, that represents the potential difference generated after administration of the visual stimulus, allowing quantification of the power of the electrical signal produced in the optic nerve. VEP trace typically consists of three waves, a first positive wave (P1), followed by both a negative (N1) and a second positive wave (P2). We measured the latency of N1 and the peak-to-peak amplitude of the N1-P2 complex to assess the optic nerve function (*Figure 1A*). While 2 and 4 months old mutant mice did not display detectable VEP alterations, 8-month-old mutant mice displayed increased latencies (*WT*: 46±4; *Wfs1*-KO: 51±2, p<0.001), indicative of slower transmissions along the visual pathway (*Figure 1A*). However, both VEP parameters were significantly affected as mouse age progressed to 12 months (VEP amplitude *WT*: 40±14; *Wfs1*-KO: 28±2; p<0.001; VEP latencies *WT*: 47±2; *Wfs1*-KO: 52±3, p<0.001) (*Figure 1A*). We then asked if these neurophysiological deficits correlate with manifested loss of visual acuity as recorded by the optomotor reflex response (OMR) based on tracking the unconditioned compensatory head movements of the animal while trying to stabilize the image of the moving environment (rotating vertical stripes). *Wfs1* mutant mice showed a progressive reduction in visual acuity with undetectable differences at 2 and 4 months of age while a significant loss was recorded at 8 months which markedly worsened 4 months later (4 months - *WT*: 0.40±0.5; *Wfs1*-KO 0.36±0.8 c/deg, 8 months - *WT*: 0.39±0.4; *Wfs1*-KO: 0.30±0.3 c/deg, p<0.05, 12 months - *WT*: 0.38±0.6; *Wfs1*-KO: 0.19±0.5 c/deg, p<0.01; *Figure 1B*). These results imply that

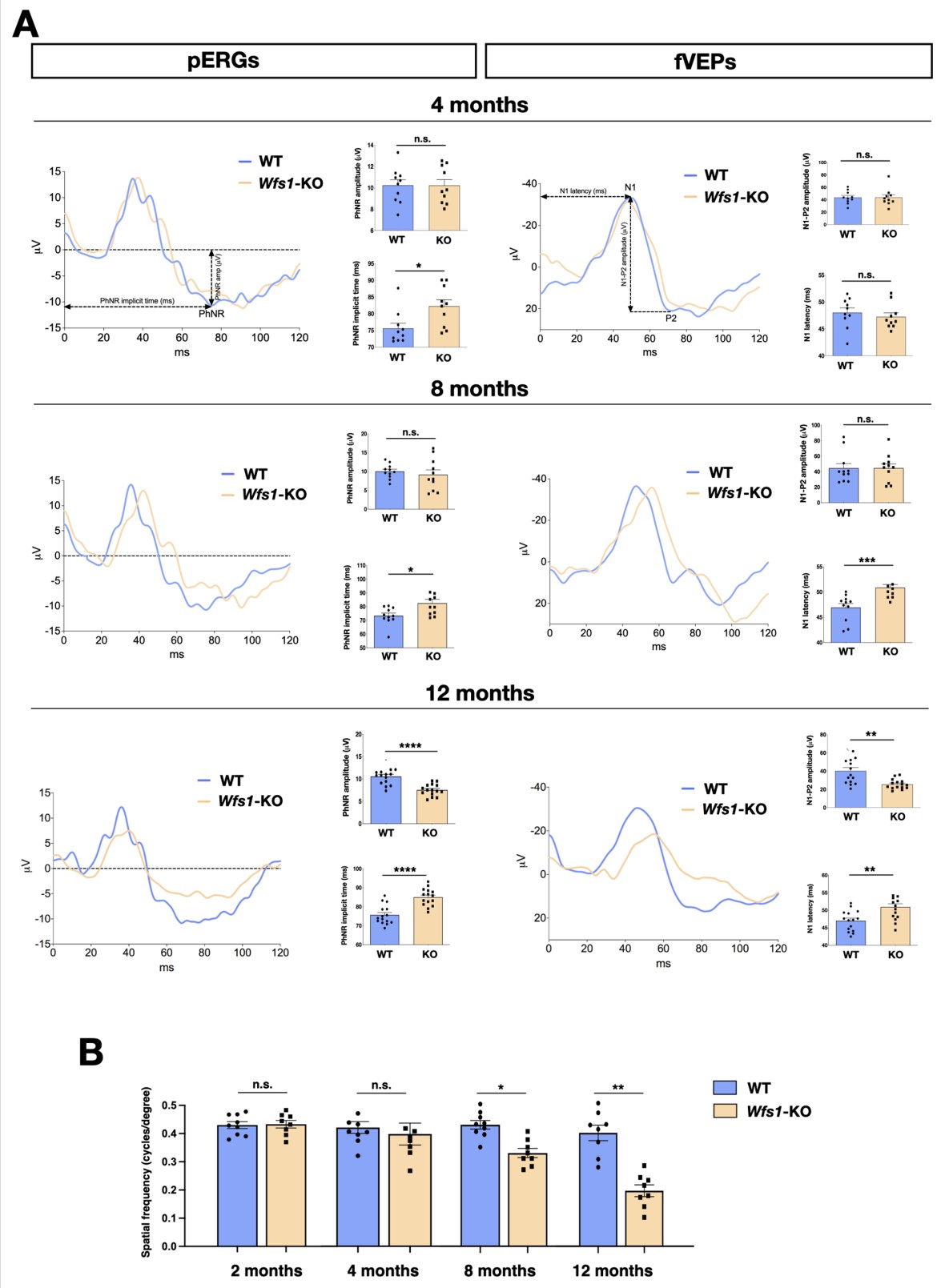

**Figure 1.** *Wfs1* mutant mice show progressive and severe impairment of visual activity. (**A**) Representative pERG and fVEP waveforms at 4, 8, and 12 months of age in wild-type (WT) and *Wfs1* mutant (KO) animals. PhNR amplitude and implicit time are quantified by ERG measurements (n=15, *p<0.05; **p<0.01; ***p<0.001, Student *t*-test). Data are presented as mean ± SEM. N1 latency and N1-P2 amplitude quantifications are calculated from the fVEP traces (n=15, *p<0.05; **p<0.01; ***p<0.001, Welch's *t*-test for N1-P2 amplitude and Student *t*-test for N1 latency). Data are presented as mean

*Figure 1 continued on next page*

Figure 1 continued

± SEM. (**B**) Quantification of visual acuity measuring the opto-motor reflex expressed as cycles per degree in 2, 4, 8, and 12 months old wild-type (WT) and *Wfs1* mutant (KO) mice (n=8, *p<0.05; **p<0.01; **p<0.001; ****p<0.0001). Data are presented as mean ± SEM.

The online version of this article includes the following figure supplement(s) for figure 1:

**Figure supplement 1.** Characterization of eye morphology in *Wfs1* mutant mice.

the initial visual impairment in *Wfs1* mutant mice is represented by RCG alterations in the retinal tissue followed by a delay of the visual stimulus along the nerve fibers suggesting a progressive optic nerve damage. Subsequently, RGC functions are extensively altered causing a drastic loss of visual acuity at 12 months of age.

## Optic nerve degeneration precedes RGC loss in *Wfs1* mutant mice

No deep characterization of the *Wfs1* mutant visual structures has been yet reported and the data available refers exclusively to eye morphology in relatively young animals (*Waszczykowska et al., 2020*). At first, we evaluated the gross retinal architecture in 8 months old mice by optical coherence tomography (OCT) and fluorescein angiography (FA), in terms of both morphology and neovascularization. FA assessment was necessary to exclude confounding effects due to diabetic retinopathy (DM), which represents one of the major eye complications of DM but whose causative mechanisms are highly different from those in WS1 (*Hilson et al., 2009*). The collected images did not reveal any signs of retinal vasculature dysfunction in either group of animals. Indeed, OCT-angiography did not highlight any significant difference in the eye fundi or perfusion and vessel length density and tortuosity (*Figure 1—figure supplement 1*). Moreover, we also assessed the thickness of retinal nerve fiber layer (RNFL), and no significant differences were disclosed by structural OCT (*Figure 1—figure supplement 1*). Since mutant elderly mice often displayed corneal haze, with consequent poor quality images, we managed to sample only three 12 months old clear-lensed eyes that still did not show big abnormalities in RNFL thickness at OCT segmentation (*Figure 1—figure supplement 1*). To investigate the cellular dysfunctions underlying the neurophysiological recordings, we performed transmission electron microscopy analyses (TEM) of transversal optic nerve sections of *Wfs1* mutant and littermate control mice at the same timepoints (8 and 12 months). Post-hoc TEM imaging analysis enabled us to estimate average RGC axonal densities (mean axon count/area), average myelin area/field also quantifying the empty space (i.e. uncovered by axons) and g-ratio (axonal area divided by axonal area including that of myelin sheets). These elaborations revealed that 8 months old *Wfs1* mutant optic nerves presented incipient myelin thinning (g-ratio - WT: 0.52±0.18; *Wfs1*-KO: 0.54±0.14, p<0.05) with reduced myelin area and a higher proportion of empty space, but without axonal count discrepancies (*Figure 2A*). TEM imaging in elderly mutant animals (12 months of age) unveiled signs of more severe myelin degeneration (g-ratio - WT: 0.42±0.18; *Wfs1*-KO: 0.50±0,18, p<0.05) with pathological features and degenerating fibers (*Figure 2B*). In particular, we observed a significant reduction of axonal density and area occupied by myelin together with evident signs of axonal damage with increased g-ratio (WT: 0.41±0.04; *Wfs1*-KO: 0.51±0.003, p<0.05; *Figure 2B*). These results provide evidence that myelin alterations and axonal thinning are the first pathological signs followed by manifested myelin degeneration and axonal loss in *Wfs1* mutant optic nerves. This pathological progression is perfectly matching the timeline of the visual functional deficits previously uncovered by electrophysiological recordings and the optomotor analysis. To assess whether optic axonal damage was occurring with simultaneous alterations in RGC cell bodies within the retinal tissue, we performed a detailed count of RGC number by immunocytochemical staining using the pan-RGCs marker RBMPS (*Figure 2C*). Unbiased stereology counting did not reveal significant RGC number changes in 8 months old *Wfs1* mutant mice (*Figure 2C*). However, 4 months later the loss of the RGC population became evident and statistically relevant (28% ± 6% loss in *Wfs1*-KO vs WT retinas) indicating a progressive loss of RGC somata over time (*Figure 2C*). These results illustrate that axonal damage and myelin degeneration anticipate the frank loss of RGC cell bodies in the retinas, even if within a relative delayed time frame a severe loss of RGCs can be also detectable in the *Wfs1*-deficient retinas.

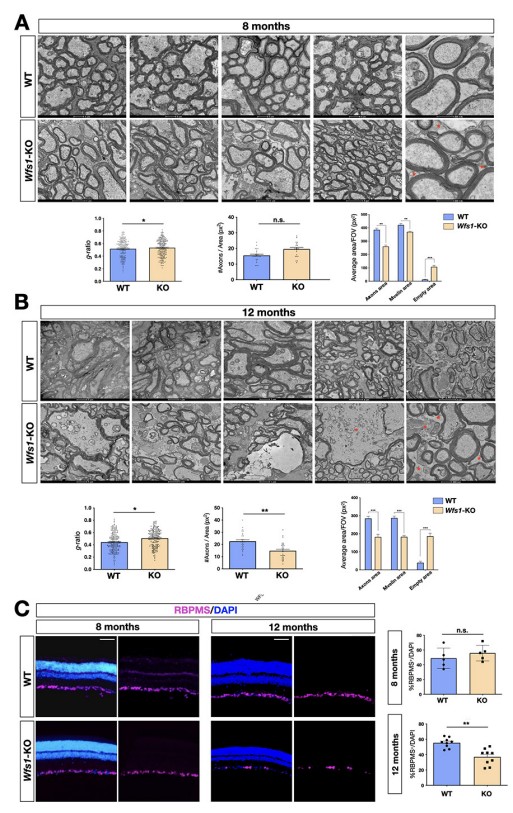

**Figure 2.** Progressive demyelination in the *Wfs1* mutant optic nerves. (**A**) Representative EM images of optic nerve cross-sections of wild-type (WT) and *Wfs1* mutant mice at 8 months of age showing increased space among the axons (red asterisks). Quantifications of the g-ratio (n=193, *p<0.05, unpaired t-test), total number of axons (n=34), myelin (n=29) and empty areas (n=29) normalized by total area (n=29) (**p<0.01; ***p<0.001, two-way ANOVA with Bonferroni's post-hoc test). Data are presented as mean ± SEM. (**B**) Representative EM images of optic nerve cross-sections of wild-type (WT) and *Wfs1* mutant mice at 12 months of age showing dramatic expansion of space among the axons (red asterisks). Quantifications of the g-ratio (n=192, *p<0.05, unpaired t-test), total number of axons (n=29), myelin (n=27) and empty areas (n=27) normalized by total area (n=27) (***p<0.001, two-way ANOVA with Bonferroni's post-hoc test). Data are presented as mean ± SEM. (**C**) Representative immunohistochemical images of retinal cross-sections of wild-type (WT) and *Wfs1* mutant (KO) animals at 8 and 12 months of age with RBPMS and DAPI staining in purple and blue, respectively. Relative quantification of RBPMS positive cells over total cell number in the GCL at 8 months (n=9, p=0.39, two-tails unpaired t-test) and at 12 months (n=8, **p<0.01, two-tails unpaired *t*-test). Scale bar: 50 μm.

# Transcriptomic analysis reveals aberrant gliosis which precedes neuronal loss in *Wfs1* mutant retinas

To unveil specific molecular alterations underlying the pathological deficits caused by the *Wfs1* gene loss, we performed transcriptome analysis of isolated mutant and control mouse retinas. To investigate prodromic events that anticipate the morphological changes, we performed RNA-seq analysis on tissues from 8 months old mice. RNAs were extracted from whole retinas of three different mice per group (WT vs *Wfs1*-KO) and subjected to global gene expression analysis. t-SNE analysis of the RNA-seq datasets confirmed the expected closer similarity and correlation between all mutant retinas in comparison to the analogous wild-type samples (*Figure 3A*). Computational analysis identified 396 genes differentially expressed (DEGs) in mutant retinas compared to the control counterparts, with a similar proportion of either up- or downregulated genes (*Figure 3B*), suggesting not profound transcriptome alterations but rather selective changes in mutant samples at this stage. Interestingly, gene enrichment analysis (Gene Ontology, GO) revealed many deregulated genes related to pathways like cell death, UPR response and protein folding (*Figure 3C*). Intriguingly, all these pathways were down-regulated in the mutant retinas, in line with the general assumption from literature that wolframin loss impairs the cellular stress-coping capacities. On the contrary, mutant retinas upregulated key genes of neuroinflammatory pathways (*Figure 3C*) as those associated with reactive glial cells like, *Gfap*, *Lcn2*, *Edn2*, *Alpk1*, and *C4* (*Figure 3D*; *Liu et al., 2022*; *Linnerbauer et al., 2022*). We, next, performed immunocytochemical analysis confirming the aberrant activation of the ER stress effectors ATF6 (*WT*: 0.5±0.3; *Wfs1*-KO: 1.5±0.2) and CHOP (*WT*: 0.2±0.01; *Wfs1*-KO: 0.4±0.02) in mutant retinal samples (*Figure 4A*). Moreover, GFAP (8 months WT: 0.5±0.2; *Wfs1*-KO: 1.4±0.3; 12 months WT: 0.2±0.02; *Wfs1*-KO: 52±2) and Vimentin (VIM) (8 months WT: 20±2; *Wfs1*-KO: 38±5; 12 months *WT*: 2.2±0.5; *Wfs1*-KO: 4.6±0.8) staining was found strongly upregulated in mutant retinal Müller glia at both 8 and 12 months of age (*Figure 4B*). Furthermore, the glia-specific glutamine synthetase (GS) enzyme was found downregulated both at mRNA and protein levels in *Wfs1*-deficient retinas (*Figures 3C and 4D*). GS generates glutamine by assembling glutamate and ammonia and, therefore, having a critical role in preventing glutamate-dependent excitotoxicity and ammonia toxicity (*Rose et al., 2013*). Thus, GS

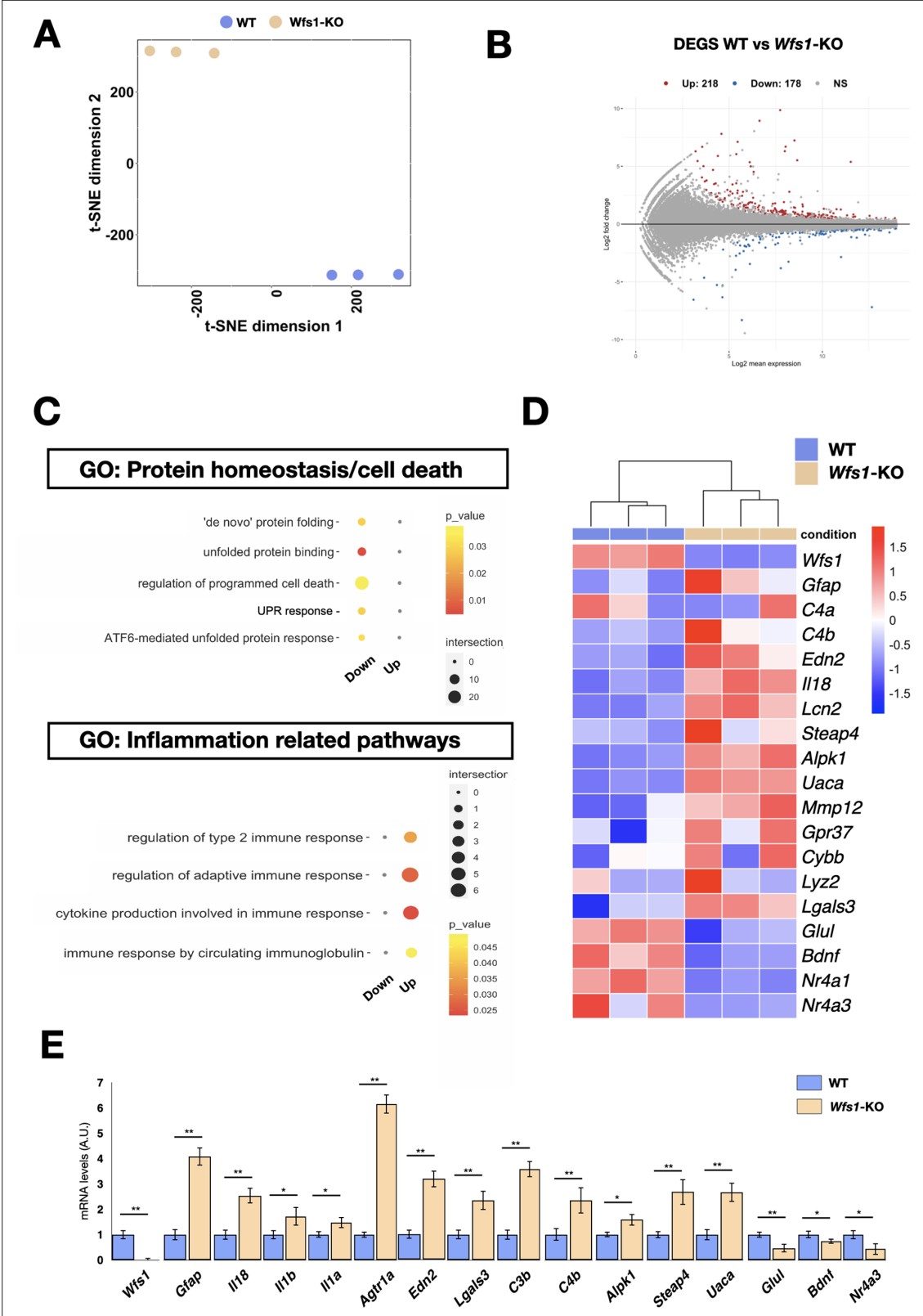

**Figure 3.** Transcriptome analysis of *Wfs1* mutant and control retinas. (**A**) Whole-transcriptome analysis using the t-Distributed Stochastic Neighbor Embedding (t-SNE) analysis with a view of the sample distribution along the first two dimensions. (**B**) Significantly up- (red) and down- (blue) regulated genes among the transcriptomic profiles of the samples, shown as highlighted dots in the MA plots. Number of differentially expressed genes (DEGs) (up = 218; down = 178). (**C**) GO-term categories relative to transcriptional analysis, as in RNA-seq dataset calculated in the list of genes downregulated

*Figure 3 continued on next page*

*Figure 3 continued*

in *Wfs1* mutant retinas. (**D**) Heatmap showing genes normalized count RPKM associated with inflammatory pathways. (**E**) Boxplot depicting the distribution of gene expression levels, confirming their trends in RNAseq. (n=3 *p<0.05; **p<0.01; ***p<0.001, Student *t*-test).

reduction in *Wfs1*-deficient retinal glial cells might favor a toxic environment detrimental for neuronal survival. Next, the genes encoding for the key neurotrophic molecule BDNF, and the survival factors Nr4a1 and Nr4a3 were also reduced in *Wfs1* mutant retinas (*Figure 3D*), further altering neuronal homeostasis and stress resilience. In fact, BDNF is a well-known neurotrophic factor expressed by RGCs and Müller glia in adulthood retina, where it exerts its autocrine or paracrine effects. Multiple lines of evidence suggest that BDNF promotes survival of RGCs and can ameliorates RGC death after traumatic optic nerve injury (*Bennett et al., 1999*). It is thus not surprising that its downregulation occurs before ganglion cell death. Finally, a number of microglial pro-inflammatory genes such as *Il-18*, *Gpr37*, *Cybb*, *Lyz2*, *Lglas3,* and *Mpp12* were upregulated, suggesting that *Wfs1* mutant microglia activate at least some reactive-specific pathways (*Figure 3D and E*). However, levels of interferon-inducible genes and NF-κB signaling components were unchanged suggesting the lack of a generalized inflammatory response, but rather a selective activation of specific neuro-inflammatory pathways. To further strengthen our results, RT-qPCR analysis were performed on independent biological samples and substantially confirmed the transcriptome data for the key selected genes (*Figure 3E*), further highlighting a selective pro-inflammatory profile in *Wfs1*-deficient retinas. On this line, we noted that the dysregulated *Lcn2* and *Vim* genes in *Wsf1* mutant samples have been shown to be direct targets of the STAT3 signaling (*Herrmann et al., 2008*). Intriguingly, previous studies reported that STAT3 can be stimulated by the ER stress-dependent PERK and IRE1α pathways in glial and cancer cells (*Meares et al., 2014*; *Chen and Zhang, 2017*). Thus, we profiled by western blotting both total and active (pY705) forms of STAT3 in control and *Wfs1* mutant retinal lysates at 8 months of age. Interestingly, the active STAT3 variant was highly enhanced in mutant samples although with some variability among the samples (*Figure 5A*). Thus, we profiled by candidate gene expression other STAT3 regulated genes such as *Fgf2*, *Il6*, *Vegfa*, *C1s,* and *C1rl* finding them consistently upregulated in *Wfs1* mutant retinas (*Figure 5B*). Altogether, these findings uncovered a prominent pro-inflammatory molecular program associated with reduction of survival and homeostatic factors which is promoted at least in part by the ER stress-dependent STAT3 activation.

## Proteomics analysis of *Wfs1* mutant retinas

Given that relative few genes showed altered expression levels in *Wfs1* mutant retinas, we postulated that additional key changes can be determined only at protein level. In fact, wolframin regulates the function of several proteins that control ER physiology and ER-mitochondrial interactions through post-transcriptional mechanisms. Thus, we collected eyes from WT and *Wfs1* mutant mice at 8 months and isolated the retinas with their optic nerves to carry out label-free LC-MS comparative proteomics (*Figure 6A*). Three biological replicates for each of the two genotypes were analysed in triplicate and aligned through Multidimensional Algorithm Protein Mapping (*Comunian et al., 2011*). Overall, 2046 proteins were identified (*Supplementary file 1*) and the virtual 2D map showed the ability of the shotgun approach to identify proteins in a wide range of molecular weight and isoelectrical point (*Figure 6—figure supplement 1*). In this study, a label-free quantification was applied, considering the total number of peptides (evaluated as Peptide Spectrum Matches or Spectral Count) for each protein. The quantitative analysis, combined with statistical evaluation, led to the identification of 21 differentially expressed proteins between the two genotypes with 9 upregulated and 12 downregulated in the *Wfs1*-KO compared to the WT condition (*Figure 6B* and *Supplementary file 1*). Notably, the most upregulated protein upon *Wfs1* gene loss was GFAP, confirming increased gliosis described earlier in the mutant retinas (*Figure 6B*). Conversely, among the most downregulated molecules in mutant samples, we identified MCT1 (Slc16a1) and basigin. MCT1 belongs to the family of monocarboxylate transporters (MCTs) that promotes the transport across cell membranes and the shuttling between cells of energy metabolites as lactate, pyruvate or ketone bodies (*Felmlee et al., 2020*; *Bosshart et al., 2021*). Basigin (known also as CD147) is a glycoprotein which assembles with MCT1 promoting its trafficking from ER and its stabilization on the cell membrane (*Kirk et al., 2000*). In brain, MCTs govern the extracellular release of lactate from glycolytic astrocytes and its transfer into neurons where it is oxidized to generate ATP (*Brooks, 2018*; *Magistretti and Allaman, 2018*). This

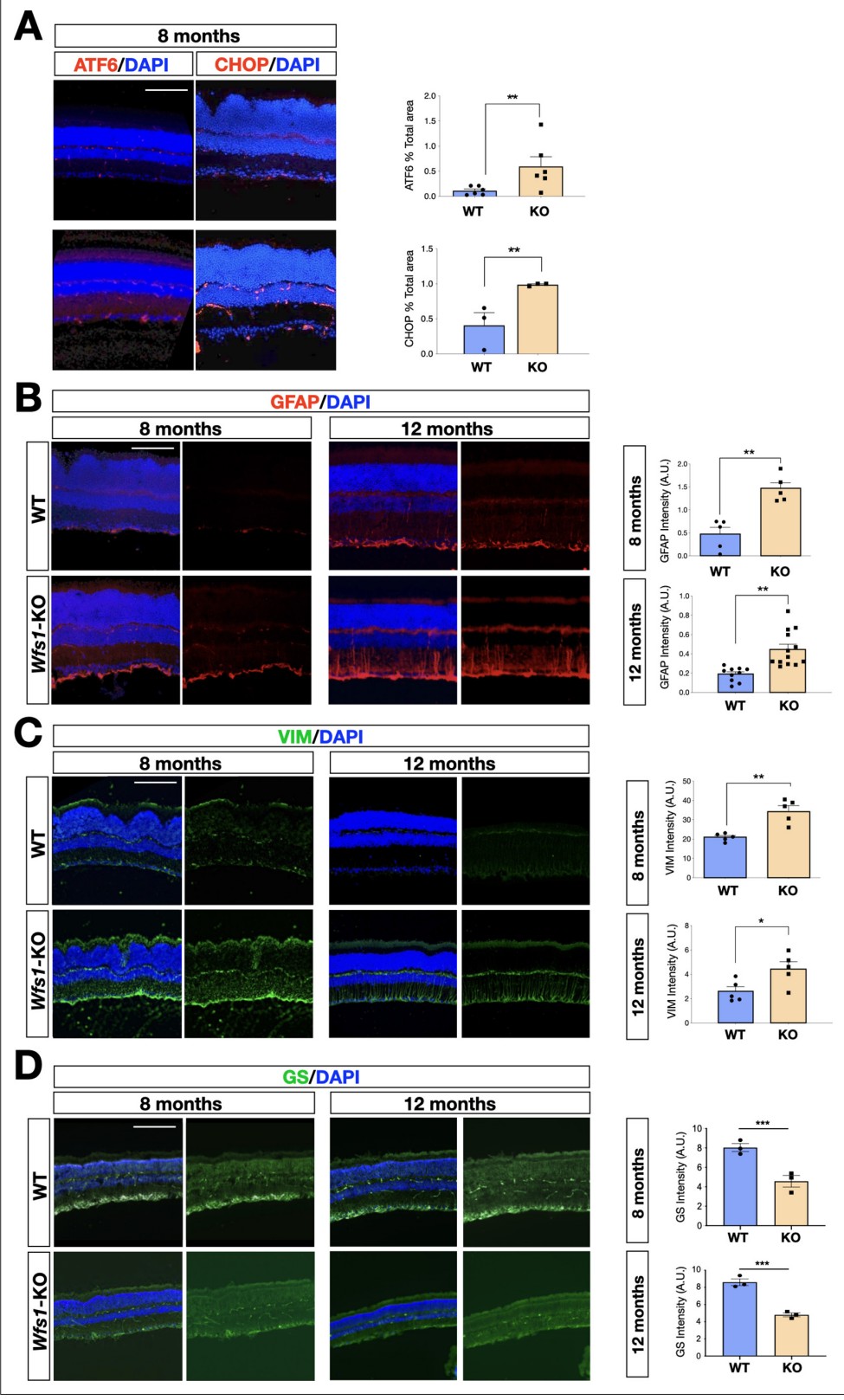

**Figure 4.** Representative immunohistochemical analysis in wild-type and *Wfs1* mutant retinal tissues at 8 and 12 months of age. (**A**) Representative images and relative quantification of ATF6 and CHOP immunofluorescence signal in wild-type (WT) and *Wfs1* mutant (KO) retinas at 8 months of age. On the right, quantification of the signal (n=5, ATF6; n=12, CHOP, **p<0.01, Student *t*-test). Data are presented as mean ± SEM. Scale bar:

*Figure 4 continued on next page*

*Figure 4 continued*

100 μm. (**B**) Representative images of GFAP immunofluorescence in 8- and 12-month-old retinas. On the right, quantification of the signal (n=5, 8 months; n=13, 12 months, \*\*p<0.01, Student *t*-test). Data are presented as mean ± SEM. Scale bar: 100 μm. (**C**) Representative images of Vimentin (VIM) immunofluorescence in 8- and 12-month-old retinas. On the right, quantification of the signal intensity of VIM in wild-type (WT) and *Wfs1* mutant (KO) retinas. (n=5, \*p<0,05, \*\*p<0.01, Student *t*-test). Data are presented as mean ± SEM. Scale bar: 100 μm. (**D**) Representative images and relative quantification of the glutamine synthetase (GS) immunofluorescence signal in wild-type (WT) and *Wfs1* mutant (KO) retinas at 8 and 12 months. Data are presented as mean ± SEM. (n=3, \*\*p<0.01, Student *t*-test). Scale bar: 100 μm.

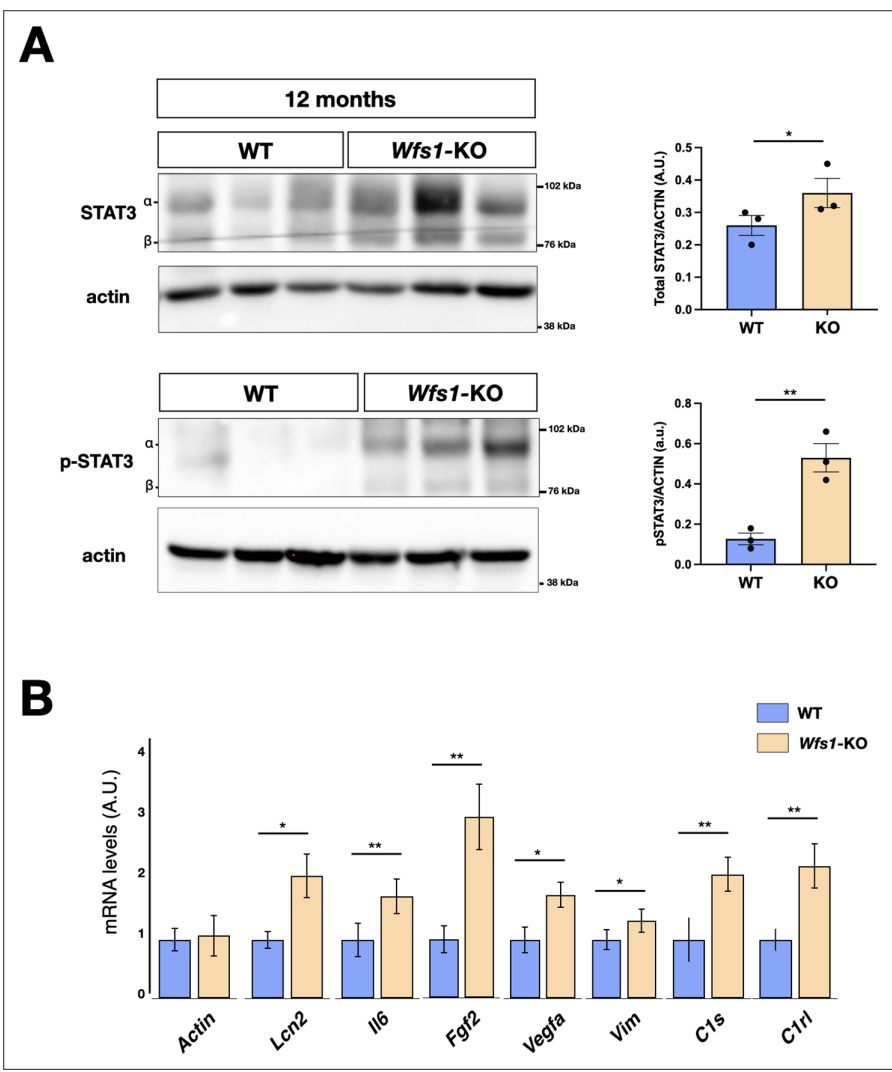

**Figure 5.** Impaired signal transducer and activator of transcription 3 (STAT3) signaling in *Wfs1* mutant mice. (**A**) Representative immunoblotting of pSTAT3 and total STAT3 endogenous protein levels in adult retinae from WT and *Wfs1* KO mice at 12 months of age. The ratio of p-STAT3 versus total signal transducer and activator of transcription 3 (STAT3) is shown in the graph after normalization over the respective protein load (actin). Quantification of pSTAT3/STAT3 protein signal intensity was performed in 12 months mice retinae (eyes from n=3 animals, \*p<0.05; \*\*p<0.01, Student *t*-test). Data are presented as mean ± SEM. (**B**) Boxplot depicting the distribution of expression levels of STAT3 regulated genes (p<0.05; \*\*p<0.01; \*\*\*p<0.001, Student *t*-test).

The online version of this article includes the following source data for figure 5:

**Source data 1.** Original western blot images used to make *Figure 5A*.

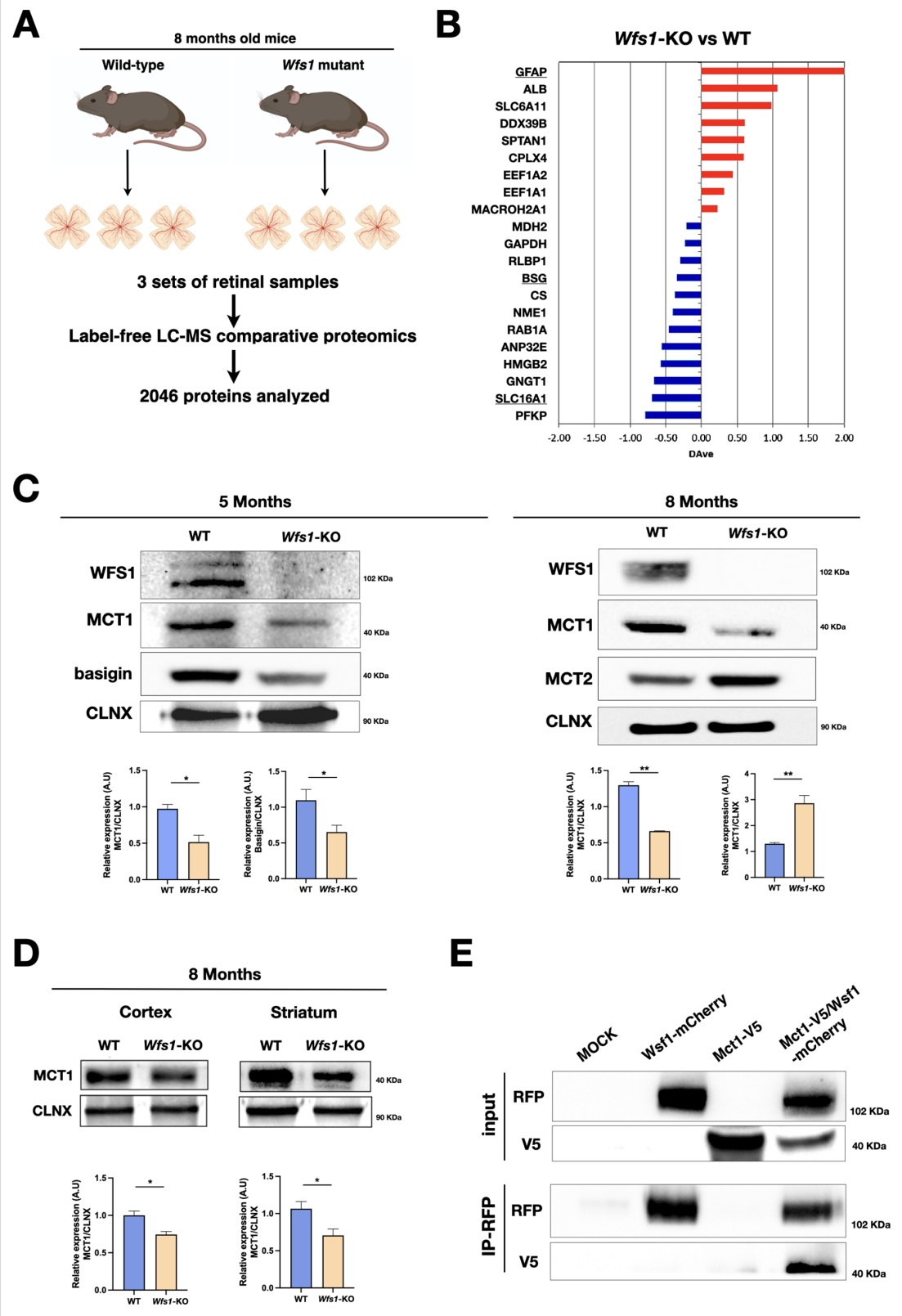

**Figure 6.** *Wfs1* mutant retinas show altered MCT1, MCT2 and basigin protein levels. (**A**) Schematic representation of the label-free LC-MS comparative proteomics approach used to analyze three set of retinae tissue samples from WT and *Wfs1* mutant mice at 8 months of age. Proteomics analysis of biological replicates for each of biological conditions identified 2046 total proteins. (**B**) Differentially expressed proteins comparing wild-type (WT) and *Wfs1* mutant (KO) mice at 8 months with p-value more significant than 0.05. The positive/red and negative/blue values indicate proteins with

*Figure 6 continued on next page*

*Figure 6 continued*

levels higher in KO or WT conditions, respectively. (**C**) Representative Western blot images and analysis of basigin, MCT1 and MCT2 signals in 5- and 8 months mice WT and *Wsf1* KO retinas with their relative quantifications. Calnexin (CLNX) was used as loading control. Data are presented as mean ± SEM (Student t-test, *p<0.05; **p<0.01). (**D**) Representative immunoblots and their relative quantifications for MCT1 protein levels in brain cortical and striatal samples in 8 months old wild-type (WT) and *Wfs1*-KO animals. Calnexin (CLNX) was used as loading control. Data are presented as mean ± SEM (Student *t*-test, *p<0.05). (**E**) Representative IP analysis from HEK293T transfected cells with the indicated constructs: *Wfs1*-mCherry and *Mct1*-V5. RFP-immunoprecipitation was performed, and blot revealed for RFP and V5, showing the proposed interaction between wolframin and MCT1 in co-transfected condition.

The online version of this article includes the following source data and figure supplement(s) for figure 6:

**Source data 1.** Original western blot images used to make *Figure 6C*.

**Source data 2.** Original western blot images used to make *Figure 6D*.

**Source data 3.** Original western blot images used to make *Figure 6E*.

**Figure supplement 1.** General analysis of the proteomics output and identified proteins.

**Figure supplement 2.** MCT1 protein loss in *WFS1* mutant HeLa cell clones.

**Figure supplement 2—source data 1.** Original western blot images used to make *Figure 6—figure supplement 2C*.

astrocyte-to-neuron lactate shuttle strengthens the metabolic coupling between glial and neuronal cells providing a rapid and tunable energy substrate to sustain the highly dynamic metabolic demand of neurons. To confirm the proteomics results, western blotting for MCT1 was performed on independent retinal tissue samples confirming a strong downregulation of this transporter both at 5 and 8 months of age (*Figure 6C*). Similarly, a marked reduction of basigin was detectable in mutant retinas at 5 months of age (*Figure 6C*). In contrast, MCT2 levels were found upregulated in 8 months old mutant retinas (*Figure 6C*). Thus, loss of MCT1 and basigin is detectable months in advance respect to the frank degeneration suggesting that the pathological process proceeds with a relatively slow progression before leading to clinically relevant symptoms. Intriguingly, MCT1 loss was detectable also in brain cortical and striatal tissue samples in 8 months old *Wfs1*-deficient mice (*Figure 6D*). To further assess wolframin and MCT1 functional relationship, we chose HeLa cells that express wolframin at appreciable levels and inactivated the gene by CRISPR/Cas9 targeted indel mutations (*Figure 6— figure supplement 2*). Interestingly, MCT1 protein levels were strongly downregulated in *WFS1*-deficient HeLa cells, but not in equivalent cells that underwent the same technical procedures but without the successful inactivation of *WFS1* (*Figure 6—figure supplement 2*). These results suggest that the wolframin-dependent regulation of MCT1 protein levels is not restricted only to the retina but is a more general phenomenon. To determine the causal relationship between wolframin and MCT1, we transiently expressed *Mct1*-V5 and *Wfs1*-mCherry constructs in HEK293 and performed co-immunoprecipitation experiments. Interestingly, MCT1-V5 was efficiently retrieved from the *Wfs1*-mCherry immunoprecipitated samples indicating that the two proteins can interact together (*Figure 6E*).

## MCT1 loss causes lactate shuttle defects in *Wfs1* mutant retina and optic nerve

Since each MCT isoform has different preferential localization on neuronal or glial cells and *Wfs1* expression in retina is not precisely defined, we next exploited a recently generated scRNA-seq of adult retinal tissues (*Fadl et al., 2020*) to determine their expression patterns at single cell resolution. Interestingly, *Wfs1* and *Slc16a1* were found predominantly co-expressed in glial and photoreceptor cells (*Figure 7A*). Conversely, expression of *Slc16a7*, encoding for MCT2, was restricted to the retinal neuronal lineage similarly to its selective cell type expression in the brain (*Figure 7A*; *Pierre and Pellerin, 2005*). We also confirmed that both wolframin and MCT1 proteins were detectable in lysates of isolated optic nerves (*Figure 7B*). Immunofluorescence analysis on optic nerve sections revealed that wolframin and MCT1 were mainly localized in MBP positive oligodendrocytes wrapping the axonal fibers (*Figure 7C*, *Figure 7—figure supplement 1*). Seminal findings have shown that myelin sheaths play a key role in glial-axonal metabolic support by shuttling metabolites like lactate and pyruvate toward axons to fuel the local energy demand (*Fünfschilling et al., 2012*; *Lee et al., 2012*). This trophic support is mediated by both MCT1 and MCT2 localized into oligodendroglia and axons, respectively (*Halestrap and Price, 1999*). This process is crucial for energy homeostasis since MCT1 specific inactivation in oligodendrocytes or Schwann cells leads to late-onset axonal degeneration

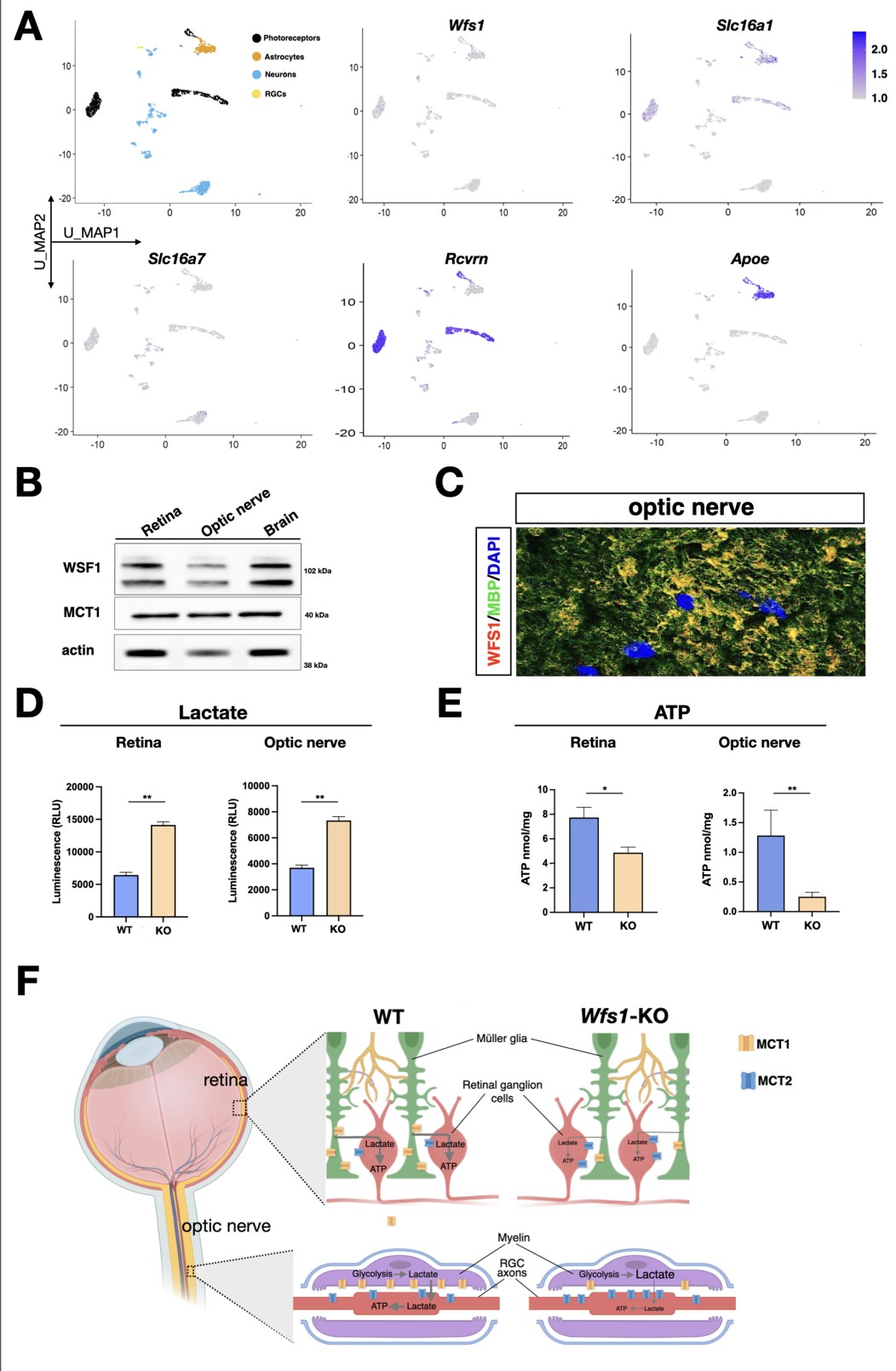

**Figure 7.** Metabolic alterations in *Wfs1* mutant retinas and optic nerves caused by MCT1 downregulation.
(**A**) Umap of the GSE153673 dataset of scRNA-seq of wild-type (WT) mouse retina showing in the first panel the main cell type composition colored by cluster followed by feature plots colored by expression level of the genes of interest (log2 expression). (**B**) Representative western blot showing wolframin and MCT1 protein levels in wild-

*Figure 7 continued on next page*

*Figure 7 continued*

type (WT) murine retinal, optic nerve, and brain lysates. Actin was used as loading control. (**C**) Representative image of colocalization of WFS1 (red) and MBP1 (green) in WT mice optic nerve section. Scale bar: 10 μm. (**D, E**) Quantification of lactate (**D**) and ATP (**E**) in WT and *Wfs1* KO mice retinae and optic nerve shows an increase of lactate level and a reduction of ATP level in *Wfs1* KO mice, respectively. Data are presented as mean ± SEM. (*p<0.05; **p<0.01, Student *t*-test). (**F**) Illustration of the key role of wolframin in maintaining physiological levels of MCT1 on glial cells to support the correct transfer of energy metabolites in retina and optic nerve. In *Wfs1* mutant retina and optic nerve, MCT1 loss results in a concomitant accumulation of lactate and loss of ATP with compensatory increased levels of MCT2 on neuronal cell bodies and axons.

The online version of this article includes the following source data and figure supplement(s) for figure 7:

**Source data 1.** Original Western blot images used to make *Figure 7B*.

**Figure supplement 1.** MCT1 protein localization in the mouse optic nerve.

---

(*Jha et al., 2020*; *Philips et al., 2021*). Thus, it is plausible that *Wfs1* gene loss might affect MCT1 protein levels in the optic nerve myelin and the transfer of lactate within the axonal tracts. If this is the case, it is expected that lactate should accumulate in oligoglial cells that are not able to metabolize themselves the large intracellular quantities. Biochemical quantifications in both retinas and isolated optic nerves showed a prominent accumulation of lactate in *Wfs1* mutant samples (*Figure 7D*). Given the failure to traffic lactate outside the oligoglial cells, both *Wfs1*-deficient cell bodies and axons of RGCs are concurrently deprived of a key energy metabolite. In fact, ATP levels were found significantly downregulated in *Wfs1* mutant optic nerve and retinal lysates (*Figure 7E*). Thus, these findings unveil a key role of wolframin in maintaining physiological levels of MCT1 on glial cells to support the correct transfer of energy metabolites both within cell bodies and axons of RGCs (*Figure 7F*).

## Discussion

Herein, we exploited unbiased transcriptomics and proteomics quantitative profiling coupled with biochemical and molecular analyses on isolated retinal tissues to disclose new mechanistic insights on RGC degeneration occurring in the WS1 mouse model. In fact, although ER stress and mitochondrial dysfunctions are unequivocally associated with *Wfs1* inactivation, they alone cannot be the cause of the specific cell type loss within the retina occurring in this disease. The retina is a complex assembly of multiple neuronal cell types and photoreceptors that are expected to be comparably sensitive to ER and mitochondrial functional impairments. Herein, we provided unprecedented findings revealing the exact molecular alterations caused by the chronic neuroinflammatory state and the pervasive bioenergetic failure occurring before the development of severe visual deficits in *Wfs1* mutant mice. Beyond the upregulation of prototype markers of inflammation in astrocytes and microglia, we also showed the downregulation of GS and BDNF that are key molecules in glial homeostasis and cell survival. GS catalyzes the conversion of ammonia and glutamate to glutamine, a precursor of glutamate and GABA, which shuttles from astrocytes to neurons (*Jayakumar and Norenberg, 2016*; *Rose et al., 2013*). Thus, GS impairment might lead to reduce clearance of extracellular brain glutamate and ammonia, glutamine deficiency, and perturbed glutamatergic and GABAergic neurotransmission. Although these findings identify a new pathological mechanism in WS1, this impairment is expected to equally affects retinal cells and to lie rather downstream to the chain of pathological events. Unbiased proteomics profiling of retinal and optic tissue samples enabled us to identify MCT1 and basigin as significantly downregulated upon *Wfs1* gene inactivation. MCT1 is the main monocarboxylate transporter in glial cells that promotes the shuttling of lactate toward neurons to sustain their elevated energy demand. Intriguingly, we confirmed a simultaneous upregulation of MCT2, the isoform predominantly localized on neuronal cell bodies and axons indicating a compensatory response of the target cells to maximize the entrance of lactate despite its poor extracellular availability (*Figure 7F*). MCT1 loss was detected both in *Wfs1* mutant retinal and optic nerve samples with concomitant accumulation of lactate and reduction of ATP. Notably, we showed that *Wfs1* expression is enriched in retinal glia and wolframin co-localizes with oligodendrocyte-forming myelin in the optic nerve confirming the co-expression with MCT1 at single-cell resolution. Among all retinal neurons, RGCs are the unique neuronal cells with myelinated axons that travel long distances within the optic nerve to reach their brain targets. Thus, these neurons are predicted to have much higher metabolic needs respect to retinal interneurons that

feature short-distance and unmyelinated projections contained within the retina. Thus, we postulate that the simultaneously loss of MCT1-dependent lactate transfer from glial cells to cell bodies and axons of RGCs represent a key pathological event which predominantly undermine the survival of this specific retinal cell type in *Wfs1* mutant animals. On this view, mice with only one *MCT1* gene copy develop axonal degeneration by 8 months of age and selective *MCT1* inactivation in oligodendrocytes causes axonal injury (*Lee et al., 2012*; *Philips et al., 2013*). Thus, these results in mouse together with observations of *MCT1* expression alterations in human neurodegenerative disease (*Lee et al., 2012*; *Andres Benito et al., 2018*; *Tang et al., 2019*), strongly point to MCT1-dependent bioenergetic loss as causative for neuronal function alterations and neurodegeneration. Although we described metabolic alterations both in retina and optic nerve in *Wfs1* mutant animals, the first detectable pathological sign is the myelin derangement in the optic nerve. These observations indicate that the optic nerve damage and axonal functional impairment likely represent the initial pathological defects caused by *Wfs1* deficiency. Remarkably, *Barboni et al., 2022* have recently described that in WS1 patients the RNFL thickness shows a fast decline since early age which precedes of about a decade the atrophy of the cellular bodies of RGCs. Thus, experimental findings in mice and patients converge in highlighting RGC axonal damage as the first disease sign in WS1. Moreover, our results uncover chronic bioenergetic failure as an unprecedented new pathological mechanism at the basis of the selective loss of RGC responsible for visual loss in WS1.

Neuroinflammation and MCT1-dependent energy insufficiency represent likely two independent phenomena triggered by separate *Wfs1* dependent pathways. However, chronic inflammation causes a loss of astrocyte-oligodendrocyte gap junctions (*Markoullis et al., 2014*). These intercellular gap junctions enable the shuttle of glucose from the blood circulation to oligodendrocytes through astrocyte intermediates (*Wasseff and Scherer, 2011*). Thus, astrocytosis can reduce glucose availability in oligodendrocytes and consequently its metabolic support to axons. On this view, MCT1 downregulation and astrocytosis create a pathological axis that synergistically impairs oligodendrocyte metabolic support capabilities.

Several hypotheses can provide an explanation of the loss of MCT1 and basigin based on the known functions of wolframin. First, wolframin was shown to direct vesicle trafficking from ER to the cell periphery and this function can sustain the delivery of specific client proteins (i.e. MCT1 and basigin) to the cell membrane. Second, wolframin can stabilize MCT1 and basigin on the membrane and therefore protect them from selective degradation. Finally, wolframin-dependent ER stress might particularly affect the stability of selective proteins. An additional hypothesis is that wolframin has a general facilitatory role in the assembly of oligomeric protein assemblies. In fact, a previous study reported the concomitant loss of both alpha and beta-subunits of the $Na^+/K^+$ ATPase suggesting a possible role of wolframin in the correct folding and assembly of multimeric complexes in ER (*Zatyka et al., 2008*). Future studies will be necessary to determine if one or more of these mechanisms are responsible for MCT1/basigin protein reduction upon wolframin loss. Our results indicate that wolframin regulates MCT1/basigin protein processing and/or stability also outside the retina causing other symptoms associated with Wolfram syndrome. In fact, MCT1 is strongly enriched in brain astrocytes and oligodendrocytes and sustain the lactate shuttle towards neuronal cells and their metabolic fitness and survival (*Fünfschilling et al., 2012*; *Lee et al., 2012*). Thus, MCT1 loss in the CNS might contribute to diffuse neurodegeneration underlying the progressive brain volume loss occurring in WS1 patients. Finally, MTC1 is the most abundantly expressed lactate transporter in peripheral nerves and its inactivation in Schwann cells leads to hypomyelination and functional deficits in sensory, but not motor, peripheral nerves (*Jha et al., 2020*). Intriguingly, WS1 patients might develop peripheral neuropathy with increased impairment of the sensory component (*Liu et al., 2006*). Future studies are warranted to determine lactate metabolism dysregulation in brain and peripheral neurons in *Wfs1* mutant mice.

These results have also profound implications on future strategies for gene therapy in WS1. In fact, given the selective RGC loss, research activities are in progress to establish gene therapy approaches to express a functional *Wfs1* gene copy in RGCs in *Wfs1* mutant mice. However, given our results we expect that this design will not efficiently prevent RGC dysfunction and subsequent loss. In contrast, *Wfs1* gene activity should be restored in glial cells of the retina and optic nerve in order to correct the trafficking of energy metabolites and sustain the bioenergetics of RGC cell bodies and axons. Altogether, by revealing new pathophysiological mechanisms underlying visual loss in *Wfs1* mutant

mice these findings uncover unprecedented therapeutic directions for WS1 calling for the exploitation of pharmacological strategies for metabolic support based on boosting glucose alternative energy substrates (*Camandola and Mattson, 2017*; *Beard et al., 2022*).

## Materials and methods
### Animals
The *Wfs1*[exon8del] targeted mouse model was a kind gift of M. Plaas and colleagues (*Luuk et al., 2008*). Mice were genotyped by multiplex PCR for assessing both WT and mutant alleles using primers *Wfs1*-KO_wtF2 5' TTGGCTTGTATTTGTCGGCC, NeoR1 5' GACCGCTATCAGGACATAGCG and *Wfs1*-KO_uniR2 5' CCCATCCTGCTCTCTGAACC (*Raud et al., 2015*). Mice were maintained on a C57BL/6 J background at San Raffaele Scientific Institute Institutional mouse facility (Milan, Italy). All procedures were performed according to protocols approved by the internal IACUC and reported to the Italian Ministry of Health according to the European Communities Council Directive 2010/63/EU.

### Optical coherence tomography analysis and fluorescein angiography
Optical coherence tomography (OCT) and fluorescein angiography (FA) were performed in collaboration with the CIS (Experimental Clinical Imaging) facility of San Raffaele Scientific Institute (Milan) as previously described (*Buccarello et al., 2017*), using the Micron IV instrument (Phoenix Research Laboratories, Pleasanton, CA, United States). Briefly, after anaesthesia, mydriasis was induced by administering a drop of tropicamide 0.5% (Visumidriatic, Tibilux Pharma, Milan, Italy) in each eye. OCT images were acquired by performing a circular scan of 550 µm of diameter around the optic nerve head. Both eyes were examined, and the results were averaged. The segmentation of retinal layers was performed using Insight software (Phoenix Research Laboratories, Pleasanton, CA, United States), OCT was followed by the FA study. A solution of 1% fluorescein (5 ml/kg Monico S.p.A., Venezia, Italy) was administered by a single intraperitoneal injection (100 µL). For each animal, the images of central and peripheral retinal vasculature were acquired.

### Photopic electroretinogram (pERG) recordings
pERGs were recorded after 10 min of light adaptation under intraperitoneal anaesthesia (80 mg/kg ketamine, 10 mg/kg xylazine). Pupils were dilated with 0.5% tropicamide and moisturized with ophthalmic gel (2% hydroxypropylmethylcellulose) to avoid eye drying. Body temperature was maintained with a homeothermic blanket system at 36.5 ± 0.5 °C. pERG was recorded from one eye at a time using a corneal electrode connected via flexible cables to a Micromed amplifier, as reported previously (*Marenna et al., 2020*). Each session included 3 trains of 10 flash stimuli (with 130 mJ intensity, 10ms duration and 0.5 Hz frequency) delivered with a flash photostimulator (Micromed, Mogliano Veneto, Italy) placed at 15 cm from the stimulated eye. pERGs were acquired with Micromed System Plus Evolution at a sampling frequency of 4096 Hz, coded with 16 bits, bandpass-filtered (5–100 Hz), and notch filtered (50 Hz). Implicit time of PhNR was measured, together with its corresponding amplitude (from baseline to negative PhNR peak). Left and right eyes were averaged to obtain single values for each animal.

### Visual evoked potential (PEV) recordings
Before recording procedures, mice were placed in a dark room and allowed to adapt to darkness for 5 min. Mice were intraperitoneally anesthetized (80 mg/kg ketamine, 10 mg/kg xylazine) and adequate level of anesthesia was verified by checking for the presence of tail-pinching reflex. Body temperature was maintained at 36.5 ± 0.5 °C by a homeothermic blanket system with a rectal thermometer probe. Both eyes were dilated with 0.5% tropicamide and protected using ophthalmic gel. Non-invasive epidermal VEPs were recorded using a 6 mm Ø Ag/AgCl cup electrode placed on the shaved scalp over V1, contralateral to the stimulated eye (1 mm anterior to interaural line and 2.5 mm contralateral to flash stimulation) and a needle electrode was inserted in the snout for reference. The cup was fixed with electro-conductive adhesive paste and at the end of first eye recording, it was placed on the opposite hemisphere to acquire VEPs from the contralateral eye, as described previously (*Marenna et al., 2019*). For each VEP recording session, 3 trains of 20 flash stimuli (with 260 mJ intensity, 10ms duration and 1 Hz frequency) were delivered with a flash photostimulator (Micromed,

Mogliano Veneto, Italy) placed at 15 cm from the stimulated eye, while the contralateral eye was covered. VEPs from both eyes were acquired and measured offline with Micromed System Plus Evolution software at a sampling frequency of 4096 Hz, coded with 16 bits, bandpass filtered (5–100 Hz) and notch filtered (50 Hz). Latency of the first negative peak (N1) and peak-to-peak amplitude (N1-P2) were measured, and then left and right eyes were averaged to obtain single values for each mouse.

## Visual acuity assessment using the optomotor reflex system

The spatial frequency threshold ('visual acuity') of the optomotor reflex index was determined using the Optodrum instrumentation (Striatech, Germany). Briefly, freely moving mice were placed on an elevated platform and exposed to vertical sine-wave gratings rotating at 12°/s. Spatial frequency of the grating at full contrast was gradually increased until the mice no longer were tracking the grating with reflexive head movements in concert with the rotation. The results are averaged for right- and left-eye acuity. Mice were tested within 5 hr of their daylight hour onset.

## Transmission electron microscopy analysis and morphometric evaluation of optic nerve fibers

The samples were fixed for 1 hr at 4 °C with 2,5% glutaraldehyde, 4% PFA in 0.1 M cacodylate buffer, pH 7.35. Fixation buffer was removed and cells washed 3 x with 0.1 M cacodylate buffer, pH 7.35 post-fixed in 2% OsO4 solution for 1 hr, and stained in 1% uranyl acetate for 1 hr at room temperature. After dehydration, specimens were embedded in Epon resin. Ultra-thin sections (about 70 nm) were stained with uranyl acetate and lead citrate and were examined by electron microscopy (Fei Talos L120CG2). For fiber counts, regions were randomly selected to minimize the bias due to fibre size distribution along the nerve. Over 200 nerve fibres (on average) were counted from individual mice (4–6 animals from each group). All nerve fibres and axon analysis were executed using AxonJ plugin of Fiji software (*Marenna et al., 2019*). Total Axon area quantifications were performed on binned and gaussian blurred images and the total was normalized per field of view area. Area based g-ratio (axon area divided by axon area plus myelin area) were used.

## Tissue collection and immunohistochemistry

Animals were euthanized and the eyes and brains collected and subjected to the different procedures based on the type of analysis. In particular after visual functional and structural analysis, mice were sacrificed and eyes with optic nerves were harvested for light and electron microscopic evaluation. Dissected retinas were fixed in 4% (w/v) PFA overnight, washed in PBS and soaked in cryoprotective solution (30% sucrose in PBS) overnight. Then, embedding in OCT Compound (VWR) was performed and eye tissues were cryo-sectioned in 18μm-thick sections and subjected to immunostaining. Where indicated, RGC density in the ganglion cell layer was determined for each eye by counting the number of cells over a 300 μm distance in serial sections representative of the full retina. For protein/RNA isolation, freshly collected retinas were flash frozen on dry-ice and stored until extraction. For all other tissues collection, mice were previously anesthetized and transcardially perfused with 0.1 M phosphate buffer (PB) at room temperature (RT) at pH 7.4. Subsequently, tissues were collected and post-fixed (for IF/IHC analysis) or flash frozen on dry ice (for RNA/Protein extraction) as needed. For immunohistochemistry, retinas were fixed in 4% of PFA, soaked in cryoprotective solution (30% sucrose in PBS), flash frozen and cut into 50 μm slices. Next, the slices were permeabilized with 3% $H_2O_2$, 10% methanol and 2% of triton for 20 min. After three washes, the slices were treated with the blocking solution (3% BSA, 0.1% tween 20) and incubated with the primary antibody over night at 4 °C. For the immunofluorescence, the secondary antibody was added, and the slices mounted for the imaging.

## Western blot analysis

Samples were homogenized in RIPA Buffer (Tris- HCl 100 mM pH 7.4, NaCl 150 mM, EGTA 1 mM, Triton 0.5%, SDS 0.1% with protease (2%) and phosphatase (10%) inhibitors (Roche)). After a 30-min incubation at 4 °C, lysates were centrifuged at 13,000 g for another 30 min and the supernatant was collected into new Eppendorf tubes. Present proteins were then quantified using the Pierce BCA Protein Assay (Thermo Scientific). Electrophoretic run of samples containing 30 μg of proteins was performed on precast NuPAGE 3–8% or 4–12% Tris-Acetate Protein Gels (Thermo Fisher Scientific), or

on home-made acrylamide gels 8–15%. After the run (30–60 min at 150 V), proteins were transferred to a nitrocellulose membrane using the Trans-Blot Turbo RTA Mini Nitrocellulose Transfer Kit (Biorad). To block the unspecific sites, membranes were incubated for 1 hr at RT in 5% non-fat dry milk (5% BSA where indicated), 0.1% PBS-Tween solution and then with the primary antibody (antibodies specifics are listed in *Supplementary file 2*) diluted in the same solution overnight at 4 °C. The day after, membranes were incubated for 1 hr at RT with the secondary antibody (Polyclonal Goat Anti-Rabbit Immunoglobulins/HRP, Dako; Polyclonal Goat Anti-Mouse Immunoglobulins/HRP, Dako) prepared in the same blocking solution at a dilution of 1:10,000. After several washes, membranes were revealed with SuperSignal West Pico PLUS Chemiluminescent Substrate ECL (Thermo Fisher Scientific). Chemiluminescence was detected using ChemiDoc Gel Imaging System (Biorad). Finally, band densitometry relative to control was calculated using ImageLab software, normalized on housekeeping as indicated in each figure (calnexin and actin).

## Co-immunoprecipitation assays

For co-immunoprecipitation (Co-IP) experiments, HEK-293T cells were cultured in 10 cm dishes. Expression vectors with *Wfs1*-mCherry and *Mct1*-V5 were used for transient transfection in cells. Cells were lysed in IP buffer (Tris-HCl pH 7.5 20 mM, NaCl 150 mM, EDTA 1 mM, EGTA 1 mM, 1% Triton X-100, Protease inhibitors and Phosphatase inhibitors in milli-Q) for 30 minutes at 4 °C, vortexing every 10 min. Then the lysed cells were centrifuged for 10 min at 14,000 g at 4 °C to pellet the debris. The supernatant was collected, and proteins were quantified using the Pierce BCA Protein Assay kit (ThermoFisher Scientific). From each sample, 60 µg were kept as the INPUT, which corresponds to the cell lysate before being co-immunoprecipitated. After that, the same amount of the remaining protein was taken from all samples (accordingly to that of the limiting sample) and brought to a final volume of 450 µL with IP buffer. Generally, not less than 2 mg were used as starting material for the subsequent IP procedure. Protein lysates were pre-cleared 1 hr at 4 °C on a rotating wheel with 50 µl of previously washed Dynabeads Protein G (Invitrogen, Thermo Fisher Scientific), to avoid aspecific bindings. The samples were then centrifuged at max speed for 10 min to precipitate the beads. The supernatant was then collected and incubated overnight at 4 °C on the rotating wheel with the antibodies anti-V5 or anti-RFP (the antibodies with their relative dilutions are listed in *Supplementary file 2*). The day after, each sample was incubated with 100 µl of washed beads for 2 hr at 4 °C on the rotating wheel, in order to precipitate the proteins bound to the antibodies. After that, using the magnetic particle concentrator (DynaMag-Spin, Invitrogen), the fluid was collected as the unbound, representing the protein fraction that was not co-immunoprecipitated. Then, still using the magnet, the beads were washed 8 times with 1 mL of IP buffer. The beads were resuspended in 100 µl of Laemmli buffer 1 X and boiled at 95 °C for 5 min, to detach the proteins from the beads. The co-immunoprecipitated fraction was separated from the beads with the magnet and collected in a new Eppendorf. The unbounds were supplemented with 100 µl of Laemmli buffer 4 X, while the INPUTs were additioned with Laemmli buffer 4 X and an appropriate amount of $H_2O$ to bring the concentration of the buffer to 1 X, reaching a final volume of 32 µl. The negative controls were obtained by immunoprecipitating the double transfected cells lysates with an isotype control (a non-specific antibody of the same species and isotype of the specific antibody used for the Co-IP).

## RNA-Seq and computational analysis

For transcriptome analysis, total RNA was extracted and purified with the QIAGEN RNeasy Micro Kit from freshly enucleated eyes from 8-month-old mice (n=3 control vs n=3 *Wfs1*-deficient mice). Sequencing was performed by GENEWIZ sequencing company (Germany). FASTQ reads were quality checked with FastQC (v 0.11.9) and adaptors trimmed with Trimmomatic (v 0.40) (*Bolger et al., 2014*), specifying default parameters. High-quality trimmed reads were mapped to the mm10 reference genome with STAR aligner (v 2.7.9 .a) using the latest GENCODE main annotation file. The construction of the gene count matrix was done using featureCounts (*Liao et al., 2014*) on.bam files generated by STAR, counting the reads associated to 'exons' features per gene. Differential gene expression on the normalized gene count matrix by RPKM method was performed with DESeq2 (v 3.13). DEGS (differentially expressed genes) were extracted by filtering on Pvalue adjusted (using 0.05 as threshold value), corrected by the Benjamini Hochberg test. GO enrichment analysis on DEGS was performed with Gprofiler2 (v 0.2.0) (*Love et al., 2014*); only overrepresented categories with

corrected Pvalue were kept, using the previously described method. Downstream statistics and Plotting were performed within the R (v 4.0.1) environment. Heatmaps were generated with Complex-Heatmap (v 3.0.215).

## qRT-PCR analysis

RNA was extracted using the TRI Reagent isolation system (Sigma-Aldrich) according to the manufacturer's instructions. For quantitative RT-PCR (qRT-PCR) assays, 1 µg of RNA was reverse transcribed using the ImProm-II Reverse Transcription System (Promega), thereafter qRT-PCR was performed in triplicate with custom designed oligos (*Supplementary file 3*) using the CFX96 Real-Time PCR Detection System (Bio-Rad, USA). using the Titan HotTaq EvaGreen qPCR Mix (BIOATLAS). Obtained cDNA was diluted 1:10 and was amplified in a 16 µl reaction mixture containing 2 µl of diluted cDNA, 1×Titan Hot Taq EvaGreen qPCR Mix (Bioatlas, Estonia) and 0.4 mM of each primer. Analysis of relative expression was performed using the ΔΔCt method, using 18 S rRNA as housekeeping gene and CFX Manager software (Bio-Rad, USA).

## Proteomics sample preparation and liquid chromatography-tandem mass spectrometry (LC-MS/MS)

Wild-type (WT) and *Wfs1* mutant retinal mouse samples isolated at 8 months were analysed in label-free proteomics approach. A total of six samples were prepared for proteomics analyses by means EasyPep Mini MS Sample Prep Kit (Thermo Fisher Scientific, Rock ford, IL, USA) for proteins extraction, reduction/alkylation and digestion by trypsin and Lys-C combination enzymes. The protein concentration was measured by Pierce BCA Protein Assay kit (Thermo Fisher Scientific, Rock ford, IL, USA). The peptides mixtures were clean-up to obtain detergent-free samples with the same kit, and then resuspended in 0.1% formic acid (Sigma-Aldrich Inc, St. Louis, MO, USA) for mass spectrometry analysis. The peptide samples were analysed on an LTQ-OrbitrapXL mass spectrometer Kit (Thermo Fisher Scientific, Rock ford, IL, USA) coupled to the Eksigent nanoLC-Ultra 2D system (AB Sciex, Dublin, CA, USA). Briefly, 0.8 µg peptides were loaded on a trap (200x500 µm ChromXP C18-CL, 3 µm, 120 Å) and eluted on a reversed-phase column (75 µm x 15 cm ChromXP C18-CL, 3 µm, 120 Å) using a acenotrile gradien;B (Eluent A: 0.1% formic acid in water; Eluent B: 0.1% formic acid in acetonitrile): 5–10% in 3 min, 10–40% in 87 min, 40–95% in 10 min, and holding at 95% B for 8 min; flow rate was 300 nL/minute. The spray capillary voltage was set at 1.7 kV, and the ion transfer capillary temperature was held at 220 °C. Full MS spectra were recorded over 400–1600 m/z range in positive ion mode, resolving power of 30,000, followed by five tandem mass spectrometry (MS/MS) on the top most intense ions selected from the full MS spectrum, using a dynamic exclusion for MS/MS analysis.

## LC-MS/MS data handling and data processing

Raw files generated were processed by Proteome Discoverer 2.1 (Thermo Fisher Scientific) using Sequest HT search engine. The experimental MS/MS spectra were compared with the theoretical mass spectra obtained by in silico digestion of a *Mus musculus* protein database containing 55282 sequences (April 2022 UniProt version; https://www.uniprot.org/). Searching criteria were used: Trypsin and Lys-C enzyme, two missing cleavage, precursor mass tolerance ±50 ppm, fragment mass tolerance ±0.8 Da. The Percolator node with a target-decoy strategy was selected to give a false discovery rate of ≤0.01 (strict) The Peptide Spectrum Matches (PSM) values of identified proteins were used for label-free quantitation approach. The whole matrix was reduced by Linear Discriminant Analysis (LDA) and a pairwise comparison (3 WT *vs* 3 *Wfs1*-KO) was performed and extracted descriptors (p-value <0.05). PSM. Differential expressed proteins were obtained by MAProma Algorithms (*Sereni et al., 2019*) and filtered by p-value (less than 0.05).

## Lactate and ATP biochemical assays

Lactate level in retinae and optic nerve was measured by the Lactate-Glo Assay (Promega). Samples were disaggregated for 20–30 s using a tissue tearor homogenizer in 50 mM Tris, pH 7.5 pre-mixed with inactivation Solution (0.6 N HCl). Then, the samples were neutralized by neutralization solution (1 M Tris base). After, 50 µl of prepared samples was added to 50 µl of working solution before reading the plate on the luminometer Mithras LB 940 (Berthold Technologies, Switzerland).

ATP concentration was determined using ATP Assay Kit (Abcam). Samples were homogenized in 100 µL of ATP Assay Buffer with a tissue Tearor. Then, the samples were centrifuged for 2–5 min at 4 °C at 13,000 g using a cold microcentrifuge to remove any insoluble material and the supernatant was collected in a new tube. After 50 µl of prepared samples was added to 50 µl of working solution before reading the optical density (OD) at OD 570 nm using the Epoch Microplate Spectrophotometer (BioTek).

### CRISPR/Cas9 gene editing in HeLa cells

We used the program CRISPOR (*http://crispor.tefor.net*) to design sgRNAs on exon 2 of the human WFS1 gene (Gene ID: 7466). Efficiency of three different sgRNAs was compared in HEK293 cells and we selected sgRNA2 (5'-ctttgaagaagtcctggaga-3') for its high activity of INDEL rate using the T7 Endonuclease I assay. Then, HeLa cells were co-transfected with the vectors U6-sgRNA2-EF1α-Blast and the pCAG-Cas9-Puro using the Lipofectamine Transfection Reagent (Invitrogen) (*Rubio et al., 2016*). Co-transfected colonies were then selected by the combination of puromycin (1 µg/ml, Sigma) and blastidicin (10 µg/ml, Thermo- Fisher Scientific) and then isolated through single colony picking. Finally, resistant cell clones were assessed by TIDE analysis followed by Sanger sequencing for isolating either clones with the targeted genomic modification in the WFS1 gene or clones with the unmodified Wfs1 gene (control cells).

### Cell line validation

HeLa and HEK-293 cells were purchased by ATCC (clones CCL-2 and CRL-1576, respectively) and their identity was confirmed by Short tandem repeat analysis. HEK-293 cells were purchased by ATCC (clone CRL-1576) and their identity was confirmed by Short tandem repeat analysis. Detection of contaminated mycoplasma in cells was performed using the MycoAlert Mycoplasma Detection Kit (Lonza) according to the manual instructions. The conditioned medium obtained from >3-day cultures did not show values over 1.2 RLU, indicating no contamination of mycoplasma.

### Statistics

All values are expressed as mean ± standard error (SEM) of at least three independent experiments, as indicated. All the statistical analysis was carried out with Prism 8 software (GraphPad software). Differences between means were analyzed using the Student's *t*-test for experimental designs with less than three groups and one variable, alternatively we applied one-way/two-way ANOVA followed by Bonferroni and Tukey post hoc test. Electrophysiological comparisons between WT and KO mice were performed using Student's t test for homoscedastic samples or Welch's t-test for heteroscedastic samples, after testing for the equality of variances with Levene's test. The null hypothesis was rejected when $p < 0.05$. In the graphs 'n.s.' indicates non-significant differences, * indicates significant differences with $p < 0.05$, ** indicates significant differences with $p < 0.01$, *** indicates significant differences with $p < 0.001$. In case of unequal variances (heteroscedasticity), Welch test was applied.

## Acknowledgements

We thank A Plaas and E Vasar for providing us the *Wfs1* mutant mice, P Podini for expert support on EM imaging and I Viganò and G Zerbini for OCT and FA analyses. We are thankful to G Frontino, L Piemonti, P Barboni and ML Cascavilla for helpful discussion. We acknowledge D Bonanomi and members of the Broccoli's lab for generous support and advice. This work was supported by a private family donation financing the work on Wolfram syndrome in VB lab at OSR and PON ELIXIR CNR-BiOMICS (PIR01_00017), Elixir Implementation Study Proteomics 2021–23 and Italian Ministry of Health (RF2019-12370396) to P.M.

## Additional information

### Funding

| Funder | Grant reference number | Author |
| --- | --- | --- |
| Italian Ministry of Health | RF2019-12370396 | PierLuigi Mauri |
| Elixir Implementation Study Proteomics 2021-23 | EISP-23-072 | PierLuigi Mauri |
| Telethon Foundation | #1350 | Vania Broccoli |

The funders had no role in study design, data collection and interpretation, or the decision to submit the work for publication.

### Author contributions

Greta Rossi, Gabriele Ordazzo, Niccolò N Vanni, Dario Di Silvestre, Letizia Bernardo, Serena G Giannelli, Sharon Muggeo, Data curation, Formal analysis, Investigation, Methodology; Valerio Castoldi, Edoardo Bellini, Data curation, Formal analysis, Validation, Investigation, Methodology; Angelo Iannielli, Data curation, Formal analysis, Validation, Investigation, Visualization, Methodology; Mirko Luoni, Supervision, Methodology; Letizia Leocani, Supervision, Validation, Methodology; PierLuigi Mauri, Formal analysis, Supervision, Validation, Investigation, Visualization, Methodology, Writing - original draft, Writing - review and editing; Vania Broccoli, Conceptualization, Resources, Data curation, Supervision, Funding acquisition, Writing - original draft, Project administration, Writing - review and editing

### Author ORCIDs

Gabriele Ordazzo http://orcid.org/0000-0002-1850-0375
Edoardo Bellini http://orcid.org/0000-0001-9766-7685
Mirko Luoni http://orcid.org/0000-0002-5006-1827
Sharon Muggeo http://orcid.org/0000-0001-7135-9780
Vania Broccoli http://orcid.org/0000-0003-4050-0926

### Decision letter and Author response

Decision letter https://doi.org/10.7554/eLife.81779.sa1
Author response https://doi.org/10.7554/eLife.81779.sa2

## Additional files

### Supplementary files

• Supplementary file 1. Master lists of all identified proteins (2046) identified in the LC-MS/MS quantitative analysis between WT and Wfs1 mutant conditions with significant Dave/DCI algorithm and p-value ($P<0.05$). Proteins with highly significant score and reported in *Figure 6B* were selected for fold change significance as determined by the Dave/DCI algorithms (>0.2 and<5, respectively; red and blue highlighted) and p-value (<0.05). Gene name is reported for protein presenting a *P*-value <0.05, only.

• Supplementary file 2. List of antibodies used in this study.

• Supplementary file 3. List of primers used in this study.

• MDAR checklist

### Data availability

Sequencing data have been deposited in GEO under accession codes GSE206150.

The following dataset was generated:

| Author(s) | Year | Dataset title | Dataset URL | Database and Identifier |
| --- | --- | --- | --- | --- |
| Broccoli V, Bellini E | 2023 | MCT1-dependent energetic failure and neuroinflammation underlie optic nerve degeneration in Wolfram syndrome mice | https://www.ncbi.nlm.nih.gov/geo/query/acc.cgi?acc=GSE206150 | NCBI Gene Expression Omnibus, GSE206150 |

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
