## [Editor Report]

The primary goal of this paper is to characterize retinal dysfunction and retinal ganglion cell degeneration in the Wfs1exon8del murine model of Wolfram Syndrome 1. The study provides fundamental insight into the timelines of degeneration as well as valuable transcriptomic and proteomic datasets. The methodologies performed are rigorous and the conclusions reached are generally well supported by the data. The results of this study are highly relevant for molecular mechanisms in Wolfram Syndrome 1 and are of potential interest to scientists interested in oligodendrocyte and neuron communication.

---

## [Decision Letter]

**Decision letter after peer review:**

Thank you for submitting your article "MCT1-dependent energetic failure and neuroinflammation underlie optic nerve degeneration in Wolfram syndrome mice" for consideration by *eLife*. Your article has been reviewed by 3 peer reviewers, one of whom is a member of our Board of Reviewing Editors, and the evaluation has been overseen by Lu Chen as the Senior Editor. The reviewers have opted to remain anonymous.

Essential revisions:

(1) Findings should be discussed relative to all relevant literature. Several key publications are missing from the manuscript. This results in several incorrect statements and failure to accurately phrase the findings of this study in the context of current knowledge. Literature omitted, includes:

Crouzier et al., Sci Transl Med 2022 characterized the impact of the loss of function in the Wfs1 knock-out animals on calcium homeostasis and mitochondrial activity in isolated cortical and hippocampal neurons.

Cairns et al., Sci Rep. 2021 and Crouzier et al., Hum Mol Genet. 2022 describe the generation and phenotyping of zebrafish models that include visual system pathology.

Chen et al., Acta Neuropathol 2022 illustrates the role of Wfs1 in Tau pathology.

(2) For glutamine synthetase findings and conclusions, Müller cells should be discussed. Müller cells perform glutamate recycling for a majority of the retina and are implicated elsewhere in the study findings.

(3) Are ERG and F-VEP measures ever "normal" in this model? "Normal" visual acuity does not translate to "normal" physiology, which is important for the interpretation of these findings.

(4) Immunohistochemical analysis illustrating MCT1 expression and localization along the optic nerve, preferably in a longitudinal section (not cross-section) so that long segments of myelinated axons can be viewed.

(5) Rephrase "protein complex" in the conclusions drawn regarding Wfs1 and MCT1 co-immunoprecipitation. These data demonstrate that these 2 proteins interact, not that they are part of a complex.

(6) Improve visibility of histological findings through higher magnification and higher resolution images.

Suggested Revisions with High Potential Impact

The 3 reviewers also agreed upon and wanted to highlight several points that could substantially increase the impact of the study. These are highlighted solely for the authors' consideration and are NOT essential revisions for resubmission.

(1) Is the downregulation of MCT1 specific to the retina or common to all neurons?

2. Is the decrease in ATP due to the decrease in lactate export or the combination of an altered mitochondria function in the RGCs plus lactate deficiency?

3. Direct demonstration of the proposed relationship between lactate accumulation and Wolframin/MCT1 depletion, i.e. in primary oligodendrocyte cultures from control and Wolframin mutant mice, would greatly strengthen this study.

4. Why is MCT2 is upregulated? Does Wfs1 also interact with MCT2?

*Reviewer #1 (Recommendations for the authors):*

1. Use of primary oligodendrocyte cultures from control and Wolframin mutant mice to directly demonstrate the proposed relationship between lactate accumulation and Wolframin/MCT1 depletion is desirable.

2. It is unclear why the authors focus on astrocytes for glutamine synthetase results and discussion. Muller cells perform glutamate recycling for a majority of the retina (including Muller cell endfeet near RGC soma and axons) and are implicated elsewhere in the study findings.

3. Please add ages and group sizes to the methods section for each assay.

4. The OCT findings in this study are contrary to published findings in this model. There is no discussion of the discrepancy.

5. ERG and f-VEP should be examined at 4 months. It is possible that visual acuity is within normal limits, but f-VEP and ERG latencies are not. OKT data indicates normal visual acuity at 4 months. However, previous literature describes thinning of the retina as early as 4 months. The ability to evaluate f-VEP and ERG data in the context of a young "baseline" would increase the clarity, significance, and potential impact of these findings.

6. Fluoromicrographs are of insufficient quality to observe the findings reported in the text. Higher magnification images of crisp focus should be included for all.

7. Optic nerve staining for oligodendrocytes should also include longitudinal sections where larger segments of myelin can be viewed along the nerve.

*Reviewer #2 (Recommendations for the authors):*

The conclusions of this paper are mostly well supported by data, but some aspects need to be addressed in order to strengthen the conclusions.

(1) The authors should analyze if the downregulation of MCT1 is also observed in other types of neurons (cortex or hippocampus for instance) to determine if the observed deficit is restricted to the retina or is a common alteration of the neurons.

(2) Several key publications are missing from the manuscript and we encourage the authors to include relevant publications related to their work. For instance, the authors claim that most of the findings on WFS1 were obtained using cell lines and patients' fibroblasts which is not correct. Indeed, a recent publication by Crouzier et al., Sci Transl Med 2022 characterized the impact of the loss of function in the Wfs1 knock-out animals on calcium homeostasis and mitochondrial activity in isolated cortical and hippocampal neurons. Moreover, the author did not cite the generation and phenotyping of zebrafish models of the pathology describing a visual phenotype (Cairns et al., Sci Rep. 2021; Crouzier et al., Hum Mol Genet. 2022). Finally, they do not cite a recent publication on the role of Wfs1 in Tau pathology (Chen et al., Acta Neuropathol 2022).

3) The function of isolated RGCs mitochondria should be analyzed in order to determine if the decrease in ATP is due to the decrease in lactate export or the combination of an altered mitochondria function in the RGCs plus lactate deficiency.

(4) An immunohistochemistry analysis of MCT1 expression should be realized on the optic nerve.

(5) The pan-RGCs marker RBMPs label only 50 % of the RGCs. Therefore, other markers should be used such as BRN3A to support the conclusion.

*Reviewer #3 (Recommendations for the authors):*

I am very convinced with the results and the conclusion made from the evidence by the authors. However, I have a few suggestions for the authors.

1. The authors completely missed to acknowledge the recent development of the zebrafish model of wolfram syndrome which shows optic atrophy, and loss of vision. In the introduction, it would be good to acknowledge the recent finding of the zebrafish model of wolfram syndrome that recapitulates the clinical hallmarks of human wolfram syndrome (Cairns et al., 2021). Cairns and colleagues have demonstrated that wfs1b knockout zebrafish has loss of vision (slower optokinetic response), thinning of optic nerve layer (OCT examination), reduced number of RGCs in the retina, and further defects in motor neurons formation.

Reference: Cairns G, Burté F, Price R, O'Connor E, Toms M, Mishra R, Moosajee M, Pyle A, Sayer JA, Yu-Wai-Man P. A mutant wfs1 zebrafish model of Wolfram syndrome manifesting visual dysfunction and developmental delay. Sci Rep. 2021 Oct 14;11(1):20491. doi: 10.1038/s41598-021-99781-0. PMID: 34650143; PMCID: PMC8516871.

2. "No deep characterization of the Wfs1 mutant visual structures has been yet reported and the data available refer exclusively to eye morphology in relatively young animals (Waszczykowska et al., 2020)"- this statement should be rewritten mentioning that this is only true for rodent model and worth acknowledging the efforts made in a zebrafish model of Wolfram syndrome.

There have been no reports of defects in eye morphology in rodents but this is not true for any wolfram syndrome model. Cairns et al., 2021 have shown the defects in eye morphology with OCT imaging, retinal thickness analysis along with behavioural characterization in the zebrafish model of wolfram syndrome. However, they have not shown the molecular mechanism that could be responsible for RGCs loss and this study shows this.

3. In the Proteomics analysis of Wfs1 mutant retinas section, "MCT1 levels were downregulated whereas the levels of MCT2 were upregulated. It is not very clear to me why the level of MCT2 is upregulated, Is it a compensatory mechanism? Does Wfs1 also interact with MCT2?

4. "To determine the causal relationship between Wolframin and MCT1, we transiently expressed MCT1-V5 and Wolframin- mCherry forms in HEK293 and performed co-immunoprecipitation experiments. Interestingly, MCT1-V5 was efficiently retrieved from the Wolframin-mCherry immunoprecipitated samples indicating that the two proteins can associate together in the same protein complex".

I am not very convinced here how the authors claim that MCT1 and Wfs1 associate in the same protein complex as it is completely possible that Wfs1 and MCT1 interact with each other and that's the reason that co-immunoprecipitate rather than assuming that they are associated in the same protein complex. They could be a part of different protein complex/pathway and interacts with each other. Additional results would be required for this statement. Have the authors done any other experiment that could suggest that MCT1 and WFS1 associate in the same protein complex?

---

## [Author Response]

Essential revisions:(1) Findings should be discussed relative to all relevant literature. Several key publications are missing from the manuscript. This results in several incorrect statements and failure to accurately phrase the findings of this study in the context of current knowledge. Literature omitted, includes:Crouzier et al., Sci Transl Med 2022 characterized the impact of the loss of function in the Wfs1 knock-out animals on calcium homeostasis and mitochondrial activity in isolated cortical and hippocampal neurons.Cairns et al., Sci Rep. 2021 and Crouzier et al., Hum Mol Genet. 2022 describe the generation and phenotyping of zebrafish models that include visual system pathology.Chen et al., Acta Neuropathol 2022 illustrates the role of Wfs1 in Tau pathology.

Following the reviewer’s suggestion, we added in the Introduction a short description on the main findings by Crouzier et al. (Sci Transl Med 2022) and on the studies about the zebrafish wfs1a/b mutants (Cairns et al., Sci Rep. 2021 and Crouzier et al., Hum Mol Genet. 2022) to complete a summary on the available WS1 animal models and their significance for the human pathology. We also cited the work of Chen et al. (Acta Neuropathol 2022) suggesting an association between wolframin loss and Tau protein aggregation and relative pathology in mutant mice.

(2) For glutamine synthetase findings and conclusions, Müller cells should be discussed. Müller cells perform glutamate recycling for a majority of the retina and are implicated elsewhere in the study findings.

As highlighted by the reviewer, our data indeed suggest that Wfs1 is equally expressed in the retinal glial cells including Muller glia and astrocytes. Thus, we revised the manuscript referring to retinal glial cells when describing the expression and function of wolframin in controlling MCT1 protein levels and its effects on RGC survival.

(3) Are ERG and F-VEP measures ever "normal" in this model? "Normal" visual acuity does not translate to "normal" physiology, which is important for the interpretation of these findings.

To answer to this reviewer’s question, we executed additional pERG and fVEP recordings in 2 and 4 months old wild-type and *Wfs1* mutant mice. Interestingly, both measurements were equivalent between the two genotypes at 2 months suggesting that retinal functions are fully operative in early adulthood despite Wfs1 gene inactivation. However, we detected an initial prolonged implicit time in pERG recordings in 4 months old *Wfs1* deficient implying that retinal dysfunctions initiate earlier than the 8 months when we performed the initial recording in the original manuscript. These new data are presented in the revised Figure 1.

(4) Immunohistochemical analysis illustrating MCT1 expression and localization along the optic nerve, preferably in a longitudinal section (not cross-section) so that long segments of myelinated axons can be viewed.

We carried out additional immunofluorescence stainings on longitudinal frozen sections of the mouse optic nerve showing a significant co-localization between MCT1 and the oligodendrocyte specific marker MBP both at low and high magnification images (new Supplementary Figure 4).

(5) Rephrase "protein complex" in the conclusions drawn regarding Wfs1 and MCT1 co-immunoprecipitation. These data demonstrate that these 2 proteins interact, not that they are part of a complex.

We agree with this reviewer’s suggestion and changed the text accordingly.

(6) Improve visibility of histological findings through higher magnification and higher resolution images.

Following this reviewer’s suggestion, the images in Figure 4 have been re-sized and loaded at higher resolution.

Suggested Revisions with High Potential ImpactThe 3 reviewers also agreed upon and wanted to highlight several points that could substantially increase the impact of the study. These are highlighted solely for the authors' consideration and are NOT essential revisions for resubmission.(1) Is the downregulation of MCT1 specific to the retina or common to all neurons?

Prompted by this reviewer’s question, we performed Western blotting with brain cortical and striatal tissue lysates confirming a significant downregulation of MCT1 in *Wfs1* mutant samples (Figure 6D).

These results indicate that the wolframin control of MCT1 protein levels is not only restricted to retinal tissue, but it is a more general phenomenon.

2. Is the decrease in ATP due to the decrease in lactate export or the combination of an altered mitochondria function in the RGCs plus lactate deficiency?

As suggested by the reviewer, the total loss of ATP that we measured in *Wfs1* mutant retinal tissues can well be the result of both the lack of the energetic metabolite transfer from glial cells together with the altered mitochondrial functions in neuronal cells. This scenario is suggested by the mitochondrial dysfunctions in *Wfs1* mutant cells already analyzed in previous publications that we have fully described in the introduction of our manuscript.

3. Direct demonstration of the proposed relationship between lactate accumulation and Wolframin/MCT1 depletion, i.e. in primary oligodendrocyte cultures from control and Wolframin mutant mice, would greatly strengthen this study.

Unfortunately, we are currently missing in the lab the expertise to generate primary mouse oligodendrocyte cultures and establishing this procedure would have required significantly more time than the period granted for revising the work. However, to provide further evidence of the close functional interplay between wolframin and MCT1, we inactivated *WFS1* by CRISPR/Cas9 technology in HeLa cells that show appreciable levels of both wolframin and MCT1. Intriguingly, we confirmed that *WFS1* mutant HeLa cells displayed a strong reduction in MCT1 protein levels. These results are presented in the new Supplementary Figure 3.

4. Why is MCT2 is upregulated? Does Wfs1 also interact with MCT2?

The MCT2 isoform is found upregulated in *Wfs1* mutant retinal tissues (Figure 6C). However, we did not find evidence of interaction of the two proteins in co-IP assays (data not shown). Importantly, we found that MCT2 does not preferentially localizes in glial cells, but conversely in neuronal cell bodies and axons. Thus, in the discussion we postulated that MCT2 is upregulated in *Wfs1* mutant retinal tissues as a compensatory mechanism in the neuronal cells to maximize the entrance of lactate despite its poor extracellular availability from the surrounding glial cells (Figure 7).